# RED-HDP-HMM: Observation-Dependent Durations for Bayesian Nonparametric Sequential Models

**Mikołaj Słupiński** [1]   **Piotr Lipiński** [1]

## Abstract

The Hierarchical Dirichlet Process Hidden Markov Model (HDP-HMM) is a Bayesian nonparametric extension of the classical Hidden Markov Model, well-suited for learning from (spatio-)temporal data. To relax the restrictive geometric assumption on state durations, the HDP Hidden Semi-Markov Model was introduced. However, both models assume stationary state durations, which limits their expressive power. In this work, we extend the HDP-HMM framework by incorporating recurrent explicit duration modeling, resulting in a more general and flexible model: the Recurrent Explicit Duration HDP-HMM (RED-HDP-HMM). We propose a Gibbs sampling method for efficient inference in this model. Empirical results on both synthetic and real-world segmentation tasks demonstrate that RED-HDP-HMM consistently outperforms the disentangled sticky HDP-HMM and the standard sticky HDP-HMM. We provide theoretical results on truncation error, expressiveness relative to HDP-HSMM.

## 1. Introduction

Modern time series - from neural recordings to financial markets to autonomous navigation - exhibit a fundamental characteristic that standard models fail to capture: *how long* a system remains in a particular regime depends on *what it has observed*. A neuron's firing burst duration correlates with stimulus intensity. A honeybee's waggle dance length encodes distance to food. A vehicle's lane-keeping duration varies with road curvature. Yet despite decades of progress in sequential modeling, no principled framework unifies three essential capabilities: discovering an unknown number of latent states, modeling realistic (non-geometric)

state durations, and allowing those durations to depend on observed data.

**The Problem.** We address unsupervised segmentation of time series into latent regimes when (i) the number of regimes is unknown and potentially unbounded, (ii) regime durations follow complex, non-memoryless distributions, and (iii) duration statistics depend on observation context. This problem arises across domains where temporal structure must be learned without supervision: identifying behavioral states from neural activity (Zoltowski et al., 2020), segmenting motion trajectories, and discovering operational modes in dynamical systems (Słupiński & Lipiński, 2024a; Linderman et al., 2017).

**Limitations of Existing Approaches.** Hidden Markov Models (HMMs) underpin sequential data analysis across speech recognition, bioinformatics, and finance (Baum & Petrie, 1966), but their memoryless Markov property imposes geometric duration distributions—an assumption violated in most real applications where states exhibit structured duration patterns (Dai et al., 2016). In this work, we validate our approach on behavioral (bee waggle dance), neural (C. elegans), and synthetic (NASCAR) benchmarks spanning diverse temporal regimes. Hidden Semi-Markov Models (HSMMs) address this through explicit duration variables: when a state is entered, the model samples a sojourn time and remains in that state for the sampled number of observations before transitioning (reviewed in Section 3.1). However, classical HSMMs require pre-specifying the number of states. Bayesian nonparametric extensions, particularly the HDP-HMM (Teh et al., 2006) and its sticky variants (Fox, 2009; Zhou et al., 2021), enable learning unbounded state spaces but retain geometric duration assumptions. The HDP-HSMM (Johnson & Willsky, 2013) combines explicit durations with infinite states, but treats duration distributions as observation-independent - ignoring the empirically observed phenomenon that state persistence depends on context.

Recent work on recurrent switching models (Linderman et al., 2017; Dong et al., 2020; Ansari et al., 2021; Słupiński & Lipiński, 2024a) demonstrates that conditioning switching dynamics on observations dramatically improves segmentation. However, these approaches assume *finite* state spaces, leaving a significant gap: to our knowledge, no

[1] Institute of Computer Science, Faculty of Mathematics and Computer Science, University of Wrocław, Wrocław, Poland. Correspondence to: Mikołaj Słupiński <mikolaj.slupinski@cs.uni.wroc.pl>.

*Proceedings of the 43rd International Conference on Machine Learning*, Seoul, South Korea. PMLR 306, 2026. Copyright 2026 by the author(s).

existing model provides observation-dependent duration modeling within a Bayesian nonparametric framework.

**Our Approach.** We introduce the **Recurrent Explicit Duration HDP-HMM (RED-HDP-HMM)**, the first Bayesian Nonparametric Hidden Markov Model with observation-dependent state durations. Our key insight is that duration distributions can be parameterized through Gamma-mixed Negative Binomial regression, where regression coefficients link past observations to the probability of extended state occupancy. This provides: (1) infinite support for arbitrarily long durations without truncation artifacts, (2) natural integration with the HDP-HMM framework via conjugate Bayesian inference, and (3) interpretable duration coefficients that reveal how observations influence state persistence.

**Contributions.** We make the following contributions:

1. **Model:** We propose RED-HDP-HMM, a Bayesian non-parametric HMM that enables state durations to depend on observation history while maintaining an unbounded state space - the first model to combine these capabilities (Section 4).

2. **Theory:** We establish rigorous foundations through three theorems (Section 5):
   - *Truncation error bounds* with exponential decay $O(\exp(-c'L))$, where $L$ is the number of active states and $c'$ depends on the concentration parameter—$L = 50$ yields error below $10^{-6}$ (Theorem 5.1);
   - *Strict expressiveness* over HDP-HSMM (Theorem 5.4).

3. **Algorithm:** We develop an efficient weak-limit Gibbs sampler with $\mathcal{O}(TL(L + D_{\max}))$ complexity, where $T$ is the sequence length and $D_{\max}$ the maximum duration.

4. **Experiments:** We demonstrate consistent improvements over state-of-the-art HDP-HMM variants on three benchmark types, with 4.5–10.4 percentage point accuracy gains on neural data (Section 7).

Beyond these primary contributions, our experiments validate two findings of independent interest: (i) nonloopy transitions outperform loopy variants when using infinite-support duration distributions, confirming observations from Słupiński & Lipiński (2024b); and (ii) recurrent connections improve neural activity modeling in the Bayesian nonparametric setting, extending results from Linderman et al. (2019) beyond finite-state models.

**Paper Organization.** Section 2 discusses related work. Section 3 reviews necessary background. Section 4 presents our model formulation. Section 5 establishes theoretical properties. Section 6 describes inference. Section 7 presents experiments, and Section 9 concludes.

Table 1 summarizes how RED-HDP-HMM uniquely occupies the intersection of infinite state modeling, explicit duration variables, and recurrent observation-dependence.

## 2. Related Work

**Bayesian Nonparametric HMMs.** The HDP-HMM (Teh et al., 2006) extended HMMs to unbounded state spaces using hierarchical Dirichlet processes, but suffers from rapid state switching due to implicit geometric durations. The sticky HDP-HMM (Fox, 2009) addresses this with a global self-transition bias, while the disentangled sticky HDP-HMM (Zhou et al., 2021) separates self-persistence strength from transition similarity. However, all these variants retain geometric duration constraints. The HDP-HSMM (Johnson & Willsky, 2013) introduced explicit duration modeling to the nonparametric setting, but durations remain observation-independent. RED-HDP-HMM occupies the remaining gap in this line of work by combining the HDP-HMM's unbounded state space with explicit non-geometric durations whose parameters depend on observations.

**Nested BNP Sequence Models.** Several Bayesian non-parametric sequence models discover segmental structure through nested or context-tree priors rather than explicit duration variables. The infinite Markov model (Mochihashi & Sumita, 2007) and nested Pitman-Yor language model (Mochihashi et al., 2009) use hierarchical Pitman-Yor structure for discrete sequence segmentation, while the stochastic memoizer and sequence memoizer (Wood et al., 2009; 2011) model unbounded discrete contexts. These methods are complementary to RED-HDP-HMM: they emphasize discrete language-like sequences and adaptive context length, whereas we target continuous-valued time series with interpretable latent states and observation-dependent duration variables. The block diagonal infinite HMM (Stepleton et al., 2009) also encourages persistent temporal structure by learning block-structured transition matrices; this can be viewed as an implicit, data-driven form of duration behavior, while RED-HDP-HMM models durations as first-class random variables.

**Recurrent Switching Models.** Recent work has shown that conditioning switching dynamics on observations significantly improves segmentation. Within the HMM/HSMM family, recurrent HSMMs condition transition or duration behavior on observations (Dai et al., 2016), but retain finite state spaces. In the Bayesian nonparametric setting, RS-HDP-HMM (Słupiński & Lipiński, 2025) extends the sticky HDP-HMM by making self-persistence probabilities

| Model | # States | Duration Model | Recurrent? | Observation Model | References |
|---|---|---|---|---|---|
| HMM (classic) | Finite | Implicit (Geometric) | No | Direct | Baum & Petrie (1966) |
| HSMM (explicit duration HMM) | Finite | Explicit (parametric, e.g., Poisson, Categorical) | No | Direct | Chiappa (2014) |
| HDP-HMM | Infinite | Geometric (implicit) | No | Direct | Teh et al. (2006) |
| Sticky HDP-HMM | Infinite | Geometric (implicit), biased for persistence | No | Direct | Fox et al. (2011) |
| Disentangled Sticky HDP-HMM | Infinite | Geometric (implicit), biased for persistence | No | Direct | Zhou et al. (2021) |
| RS-HDP-HMM | Infinite | Geometric (implicit), recurrent | Yes | Direct | Słupiński & Lipiński (2025) |
| HDP-HSMM | Infinite | Explicit (state-specific distribution) | No | Direct | Johnson & Willsky (2013) |
| Sticky HDP-SLDS | Infinite | Geometric (implicit), biased for persistence | No | Direct | Fox et al. (2011) |
| recurrent SLDS (rSLDS) | Finite | Implicit | Yes | Linear latent | Linderman et al. (2017) |
| REDSLDS | Finite | Explicit (recurrent, conditionally categorical) | Yes | Linear latent | Słupiński & Lipiński (2024a) |
| SNLDS | Finite | Implicit | Yes | Latent nonlinear | Dong et al. (2020) |
| REDSDS | Finite | Explicit (recurrent, conditionally categorical) | Yes | Latent nonlinear | Ansari et al. (2021) |
| EDSLDS | Finite | Explicit (parametric, e.g., Poisson, Categorical) | No | Linear latent | Słupiński & Lipiński (2024b) |
| **RED-HDP-HMM (Ours)** | **Infinite** | **Explicit (recurrent)** | **Yes** | Direct (extendable) | **This paper** |

*Table 1.* Comparison of the proposed model with related approaches. RED-HDP-HMM is the first to combine Bayesian nonparametric infinite state modeling with recurrent explicit-duration mechanisms. "Linear latent" and "nonlinear latent" indicate models with additional continuous state variables evolving under linear or nonlinear dynamics, respectively.

observation-dependent through logistic regression, using Pólya-Gamma augmentation for efficient inference. While RS-HDP-HMM introduces recurrence to the Bayesian non-parametric setting, it retains implicit geometric durations - the same limitation that motivates explicit duration modeling in our approach. A parallel line of recurrent switching state-space models, including rSLDS (Linderman et al., 2017), SNLDS (Dong et al., 2020), REDSDS (Ansari et al., 2021), and REDSLDS (Słupiński & Lipiński, 2024a), demonstrates the same observation-dependent switching principle with continuous latent dynamics, but these models assume finite state spaces.

**Duration Modeling.** Classical HSMMs (Levinson & View Profile, 1986; Chiappa, 2014) model state durations explicitly but require fixed state counts. Duration distributions range from simple geometric to Poisson, negative binomial, and finite categorical. Zhou et al. (2012) introduced the LGMNB regression model for count data, which we adapt for duration prediction. Recent work (Słupiński & Lipiński, 2024b) suggests that infinite-support distributions combined with nonloopy transitions improve segmentation quality.

Our work uniquely combines these threads: Bayesian non-parametric flexibility from HDP-HMMs, explicit duration modeling from HSMMs, and observation-dependence from recurrent switching models.

## 3. Preliminaries

### 3.1. Hidden Semi-Markov Models

*Hidden Semi-Markov Models (HSMMs)* generalize HMMs by explicitly modeling state sojourn time. When state $z$ is entered, duration $d$ is drawn from $P(D = d \mid z)$, and the system remains in state $z$ for $d$ observations before transitioning. This enables modeling of Poisson, Gamma, or other duration distributions rather than the geometric durations implicit in standard HMMs. However, traditional HSMMs require pre-specifying the number of states.

### 3.2. HDP-HMM and HDP-HSMM

The HDP-HMM (Teh et al., 2006) places Dirichlet process priors on transition distributions, enabling unbounded state spaces. The generative process uses stick-breaking weights $\boldsymbol{\beta} \sim \text{GEM}(\gamma)$ as a shared global prior, with state-specific transitions $\pi_j \sim \text{DP}(\alpha\boldsymbol{\beta})$. Latent states follow $z_t \sim \pi_{z_{t-1}}$ and observations $y_t \sim f(\theta_{z_t})$.

The HDP-HSMM (Johnson & Willsky, 2013) extends this framework with explicit duration variables:

$$\text{Global prior:} \quad \boldsymbol{\beta} \sim \text{GEM}(\gamma), \quad \bar{\pi}_i \overset{iid}{\sim} \text{DP}(\alpha\boldsymbol{\beta}), \quad (1)$$

$$\text{Parameters:} \quad (\theta_i, \omega_i) \overset{iid}{\sim} H \times G, \quad (2)$$

$$\text{Sequence:} \quad z_s \sim \pi_{z_{s-1}}, \quad d_s \sim g(\omega_{z_s}), \quad (3)$$

$$\text{Observations:} \quad y_t \sim f(\theta_{z_t}), \quad (4)$$

where $g$ is the duration distribution and self-transitions are eliminated: $\pi_{ij} = \bar{\pi}_{ij}(1 - \delta_{ij})/(1 - \bar{\pi}_{ii})$. While HDP-HSMM enables state-specific duration distributions, these remain observation-independent.

### 3.3. Negative Binomial Regression

Zhou et al. (2012) introduced Negative Binomial regression for count data:

$$d_i \sim \text{NB}(r, p_i), \quad \psi_i = \text{logit}(p_i) = \boldsymbol{y}_i^T \boldsymbol{\beta}^D + \ln \epsilon_i, \quad (5)$$

where $\text{logit}(p) = \ln(p/(1 - p))$ is the log-odds function with inverse $\sigma(x) = 1/(1 + e^{-x})$ (the logistic sigmoid), $\epsilon_i$ is a lognormal random effect, and $r$ has a gamma prior. This provides a flexible count regression model with infinite support, which we adapt for duration prediction.

## 4. Model Formulation

**Key Insight.** Our central idea is to make state durations depend on observations through regression. When the system enters a new state at time $t$, we sample a duration $d_t$ from a Negative Binomial distribution whose parameters

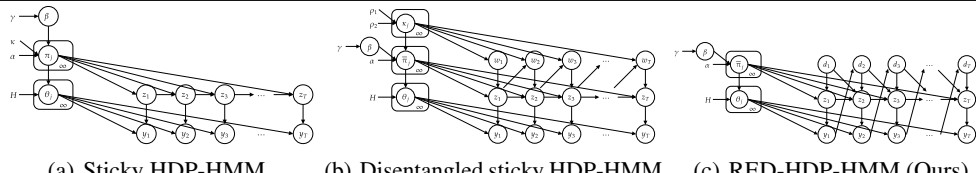

(a) Sticky HDP-HMM     (b) Disentangled sticky HDP-HMM     (c) RED-HDP-HMM (Ours)

*Figure 1.* Graphical models comparing (a) Sticky HDP-HMM, (b) Disentangled Sticky HDP-HMM, and (c) our RED-HDP-HMM. The key innovation is the arrow from $y_{t-1}$ to $d_t$, enabling observation-dependent durations.

depend on the preceding observation $y_{t-1}$. This captures the intuition that what the system has just observed should influence how long it stays in its current regime.

This extension is deliberately minimal. The Negative Binomial regression layer preserves Pólya-Gamma conjugacy, leaves the HDP transition machinery intact, and keeps duration coefficients directly interpretable. These features are what make closed-form Gibbs updates, finite-truncation analysis, and the expressiveness comparison in Section 5 possible.

**State Transitions.** Following the notation of HDP-HMM variants and recurrent models (Ansari et al., 2021; Słupiński & Lipiński, 2024a), we define state transitions as:

$$p\left(z_t \mid z_{t-1}, d_{t-1}, \mathbf{y}_{t-1}\right) = \begin{cases} \delta_{z_{t-1}} & \text{if} \quad d_{t-1} > 1 \\ \pi & \text{if} \quad d_{t-1} = 1 \end{cases}. \tag{6}$$

We define the duration distribution as

$$\begin{aligned} &p(d_t \mid z_t, d_{t-1}, \mathbf{y}_{t-1}) \\ &= \begin{cases} \delta_{d_{t-1}-1} & d_{t-1} > 1, \\ \mathrm{NB}(d_t - 1; r_{z_t}, p_{t-1}) & d_{t-1} = 1, \end{cases} \end{aligned} \tag{7}$$

where

$$\psi_{t-1} = \mathrm{logit}(p_{t-1}) = \mathbf{y}_{t-1}^T \boldsymbol{\beta}_{z_t}^D + \beta_{z_t}^{D\mathrm{bias}}. \tag{8}$$

Here $\psi_{t-1}$ is the scalar linear predictor, i.e., the log-odds of the Negative Binomial success probability $p_{t-1}$, so $p_{t-1} = \sigma(\psi_{t-1})$. The vector $\boldsymbol{\beta}_{z_t}^D \in \mathbb{R}^M$ contains state-specific regression coefficients linking observations to duration, and $\beta_{z_t}^{D\mathrm{bias}}$ is a bias term. Unlike the original LGMNB formulation (Equation 5), we omit the multiplicative error term $\varepsilon$; empirically, including this term introduces posterior multimodality that destabilizes MCMC mixing without improving segmentation accuracy.

**Autoregressive Emissions.** For spatiotemporal data where consecutive observations exhibit temporal correlations, we extend the emission model to a first-order vector autoregressive (VAR(1)) formulation:

$$\mathbf{y}_t \mid z_t = j, \mathbf{y}_{t-1} \sim \mathcal{N}(A_j \mathbf{y}_{t-1} + b_j, \Sigma_j) \tag{9}$$

where $A_j \in \mathbb{R}^{M \times M}$ is the state-specific AR coefficient matrix capturing temporal dynamics, $b_j \in \mathbb{R}^M$ is an affine bias term, and $\Sigma_j \in \mathcal{S}_{++}^M$ is the innovation covariance. This generalizes the standard emission model $\mathbf{y}_t \mid z_t = j \sim \mathcal{N}(\mu_j, \Sigma_j)$, which is recovered when $A_j = 0$ and $b_j = \mu_j$. Under stability conditions (all eigenvalues of $A_j$ satisfy $|\lambda| < 1$), the AR process within each state has a unique stationary distribution $\mathcal{N}(\mu_j^*, \Sigma_j^*)$ where $\mu_j^* = (I - A_j)^{-1} b_j$ and $\Sigma_j^*$ solves the discrete Lyapunov equation $\Sigma_j^* = A_j \Sigma_j^* A_j^\top + \Sigma_j$.

Similarly to the previous works, we take $\bar{\pi}_j \stackrel{iid}{\sim} \mathrm{DP}(\alpha \boldsymbol{\beta})$.

We test two variants of our model:

- nonloopy, where $\pi_{ij} = \frac{\bar{\pi}_{ij}(1 - \delta_{ij})}{1 - \bar{\pi}_{ii}}$,

- loopy, where $\pi = \bar{\pi}$.

In the non-loopy scenario, we lose conjugacy, necessitating the sampling of auxiliary variables $\left\{\rho_t^k\right\}_{t=1}^T$. Each of these $\rho_t^k$ is independently sampled from a geometric distribution on $\{0, 1, \ldots\}$ with a success parameter $1 - \pi_{kk}$, specifically: $\rho_t^k \mid \pi_{kk} \sim \mathrm{Geo}(1 - \pi_{kk})$.

For a detailed derivation, please refer to (Johnson & Willsky, 2013).

## 5. Theoretical Properties

In this section, we establish key theoretical properties of the RED-HDP-HMM, including truncation error bounds for the weak-limit approximation, and expressiveness relative to existing models. We write $d_{\mathrm{TV}}(P, Q) = \sup_A |P(A) - Q(A)|$ for total variation distance. Theorem 5.1 controls the state truncation level $L$, Theorem 5.2 controls the duration cap $D_{\max}$, and Corollary 5.3 combines the two independent approximation errors. These results complement finite-dimensional DP approximations and stick-breaking truncation analyses (Ishwaran & Zarepour, 2000; Ishwaran & James, 2001; Ishwaran & Zarepour, 2002b); to our knowledge, analogous finite-horizon truncation bounds with sequential error accumulation have not been stated explicitly for HDP-HMM or sticky HDP-HMM samplers (Fox et al., 2011).

**Theorem 5.1** (Truncation Error for Finite Horizons). *Let $\Pi_L^{(T)}$ denote the predictive distribution over observation sequences of length $T$ under the $L$-truncated RED-HDP-HMM, and let $\Pi_\infty^{(T)}$ denote the corresponding distribution under the full infinite-state RED-HDP-HMM prior. Define $c := \log((\gamma + 1)/\gamma)$. Then for all $L \geq 2$,*

$$d_{\mathrm{TV}}\left(\Pi_L^{(T)}, \Pi_\infty^{(T)}\right) \leq T \cdot \exp(-c(L-1)). \quad (10)$$

*Proof strategy.* Couple the full and $L$-lumped processes so they agree until the full process enters the GEM tail, then apply the expected tail-mass bound and a union bound over $T$ time steps; see Appendix .2. For $\gamma = 1$, $c = \log 2$, so $L = 50$ makes the per-step tail probability about $10^{-15}$.

**Theorem 5.2** (Duration Truncation Error Conditional on Parameters). *Let $\Theta$ denote the model parameters, including duration parameters $\{r_j, p_j(\cdot)\}$. Assume there exists $\epsilon > 0$ such that $p_j(y) \in [\epsilon, 1 - \epsilon]$ for all states $j$ and contexts $y$, and that dispersion parameters satisfy $r_j \leq R$ for some $R < \infty$. Let $\Pi^{(T)}(\cdot \mid \Theta)$ and $\Pi_{D_{\max}}^{(T)}(\cdot \mid \Theta)$ denote the distributions on length-$T$ trajectories under the full and clamped ($d' = \min(d, D_{\max})$) models, respectively. Then there exist constants $C(R, \epsilon) < \infty$ and $\lambda(\epsilon) > 0$ such that:*

$$d_{\mathrm{TV}}\big(\Pi_{D_{\max}}^{(T)}(\cdot \mid \Theta), \Pi^{(T)}(\cdot \mid \Theta)\big) \leq C(R, \epsilon)\, T\, e^{-\lambda(\epsilon) D_{\max}}. \quad (11)$$

*Proof strategy.* Use a Chernoff bound for the Negative Binomial tail, uniformly over $p_j(y) \in [\epsilon, 1 - \epsilon]$ and $r_j \leq R$, and couple the full and clamped processes so they diverge only when a sampled duration exceeds $D_{\max}$; see Appendix for details.

**Corollary 5.3** (Joint State-Duration Truncation). *For the $(L, D_{\max})$-truncated RED-HDP-HMM:*

$$d_{\mathrm{TV}}\left(\Pi_{L,D_{\max}}^{(T)}, \Pi_\infty^{(T)}\right) \leq T \cdot e^{-c(L-1)} \\ + C(R, \epsilon) \cdot T \cdot e^{-\lambda(\epsilon) D_{\max}}. \quad (12)$$

*Proof.* By the triangle inequality with intermediate distribution $\Pi_L^{(T)}$ (state-truncated, unbounded duration):

$$d_{\mathrm{TV}}\big(\Pi_{L,D_{\max}}^{(T)}, \Pi_\infty^{(T)}\big) \leq d_{\mathrm{TV}}\big(\Pi_{L,D_{\max}}^{(T)}, \Pi_L^{(T)}\big) \\ + d_{\mathrm{TV}}\big(\Pi_L^{(T)}, \Pi_\infty^{(T)}\big).$$

The second term is bounded by Theorem 5.1. For the first term, Theorem 5.2 applies to the $L$-truncated model since the duration parameters $(r_j, \beta_j^D)$ for $j \leq L$ are unchanged, and the same $\epsilon$-bound on success probabilities holds.

**Theorem 5.4** (Expressiveness with NB Durations). *The RED-HDP-HMM model class strictly contains the HDP-HSMM model class when both use Negative Binomial (NB) duration distributions. Specifically:*

1. ***Containment***: *For any HDP-HSMM with NB duration parameters $(\boldsymbol{\beta}, \{\pi_j\}, \{\theta_j\}, \{(r_j, p_j)\})$, there exists a RED-HDP-HMM with the same joint distribution over states and observations.*

2. ***Strictness***: *There exist distributions representable by RED-HDP-HMM with NB durations that cannot be represented by any HDP-HSMM with NB durations.*

*Proof strategy.* Containment follows by setting recurrent duration coefficients to zero; strictness follows because a nonconstant RED duration probability $p(y)$ induces conditional duration laws that no observation-independent HDP-HSMM duration distribution can match. The full argument is in the appendix.

Full proofs are provided in the Supplementary Material.

# 6. Inference and Learning

---

**Algorithm 1** Weak-limit sampler for RED-HDP-HMM

---

1: Jointly sample $\{z_t, d_t\}_{t=1}^T$.
2: Sample Pólya-Gamma auxiliaries $\omega_{j,t} \sim \mathrm{PG}(r_j + d_t, \psi_{j,t})$ for segment starts (see §D.5).
3: Compute $\{\psi_{j,t}\}$ for $j = 1, \ldots, L$ for $t = 1, \cdots, T, j = 1, \ldots, L$.
4: Sample $\boldsymbol{\beta}, \{\bar{\pi}_j\}_{j=1}^L$.
5: If nonloopy sequentially sample auxiliary variables $\{\rho_t^j\}$ for $t = 1, \ldots, T-1, j = 1, \ldots, L$.
6: Sample $\{\theta_j\}_{j=1}^L$.
7: Sample hyperparameter $\alpha, \gamma$ and $\{r_j, \boldsymbol{\beta}_j^D, \beta_j^{D\mathrm{bias}}, \alpha_j^D\}$ for $j = 1, \ldots, L$.

---

The weak-limit sampler for the sticky HDP-HMM exploits the fact that the Dirichlet process is inherently discrete, enabling the creation of a finite approximation of the HDP prior. This approximation tends toward the HDP prior as the number of components, $L$, approaches infinity (Ishwaran & Zarepour, 2000; 2002a). Utilizing this approximation, the conventional HMM forward-backward algorithm can be employed to sample latent variables $\{z_t\}_{t=1}^T$, thereby enhancing the Gibbs sampler's mixing rate. Our weak-limit Gibbs sampler draws from the methodology used in the DS-HDP-HMM weak-limit sampler, allowing it to simultaneously sample pairs $\{z_t, d_t\}_{t=1}^T$, akin to how DS-HDP-HMM samples sets $\{z_t, w_t\}_{t=1}^T$ simultaneously.

In Step 1, the forward-backward method is employed to simultaneously sample the two-dimensional latent variables $\{z_t, d_t\}_{t=1}^T$. A straightforward implementation of this would involve a computationally intensive $\mathcal{O}(TL^2 D_{\max}^2)$

process, which is inefficient for large $D_{\max}$. However, by leveraging the characteristic that duration variables either decrease by 1 or reset each timestep, we can pre-calculate certain components in the forward-backward equations. This optimization leads to an algorithm with complexity $\mathcal{O}(TL(L + D_{\max}))$.

Step 2 involves sampling $TL$ auxiliary variables, requiring $\mathcal{O}(TL)$ steps.

Step 3 involves computing $TL$ variables, requiring $\mathcal{O}(TL)$ steps.

Step 4 involves sampling $TL$ variables, requiring $\mathcal{O}(TL)$ steps.

In Step 5, we draw samples of $\boldsymbol{\beta}$ and $\bar{\pi}$ employing Dirichlet-categorical conjugacy, utilizing the auxiliary variables $\{m_{jk}\}_{j,k=1}^{L}$, the empirical transition matrix $\{n_{jk}\}_{j,k=1}^{L}$, and the approximate prior.

In Step 6, a conjugate prior is assigned to $\theta_j$, and conjugacy is utilized to draw samples from the posterior.

The entire process yields an algorithm with $\mathcal{O}(TL(L + D_{\max}))$ complexity, primarily influenced by the forward-backward step. For the S-HDP-HMM and DS-HDP-HMM models, the weak-limit samplers exhibit a complexity of $\mathcal{O}(TL^2)$, which is also mainly determined by the forward-backward pass. For a comprehensive derivation of weak-limit sampling approaches, please consult (Fox et al., 2011) or the supplementary materials of (Zhou et al., 2021).

# 7. Experiments

We evaluate RED-HDP-HMM on three benchmark types: synthetic NASCAR data (Linderman et al., 2017), behavioral Bee Dance data (Oh et al., 2008), and neural activity recordings from C. elegans (Kato et al., 2015). All experiments use purely self-supervised segmentation without labeled training data. These benchmarks are standard in the switching models literature (Linderman et al., 2017; Nassar et al., 2019; Ansari et al., 2021; Zhou et al., 2021; Yezerets et al., 2024).

Source code is available at https://github.com/CIRGLaboratory/RED-HDP-HMM.

**Baselines.** We compare against S-HDP-HMM (Fox et al., 2011), DS-HDP-HMM (Zhou et al., 2021), HDP-HSMM (Johnson & Willsky, 2013) and RS-HDP-HMM (Słupiński & Lipiński, 2025), the state-of-the-art Bayesian nonparametric models for self-supervised time series segmentation.

**Emission Model.** All experiments use the AR(1) emission model in Equation 9, not a memoryless Gaussian mixture. Each state has its own dynamics matrix $A_j$, affine bias $b_j$,

and innovation covariance $\Sigma_j$, so the model is a switching vector autoregression. The standard Gaussian emission is recovered only in the special case $A_j = 0$. This choice is natural for spatiotemporal data where consecutive observations exhibit temporal correlations, while the MNIW prior on $(A_j, b_j, \Sigma_j)$ preserves conjugacy for efficient Gibbs sampling (see Appendix D for prior specifications). Higher-order VAR($p$) emissions can be handled by stacking lagged observations in the regression design; nonlinear or neural emissions are possible extensions but would require non-conjugate inference, such as variational or particle-based methods.

**Model Variants.** We examine two variants: *loopy* (self-transitions permitted) and *nonloopy* (state must change when duration expires). Results in Table 2 show nonloopy consistently outperforms loopy, confirming that infinite-support duration distributions pair better with nonloopy transitions (Słupiński & Lipiński, 2024b). The nonloopy variant also removes duration-transition aliasing: persistent sojourns cannot be explained by repeated self-transitions and must be captured by the explicit duration mechanism.

**Truncation Levels.** We use truncation level $L = 50$ states and maximum duration $D_{\max} = 200$ for all experiments. These values were chosen as a single conservative setting that makes the truncation cap practically inactive for the datasets considered, while keeping the $\mathcal{O}(TL(L + D_{\max}))$ sampler computationally manageable. By Theorems 5.1 and 5.2, these settings yield truncation error below $10^{-6}$ for sequences up to $T \sim 10^4$, which is conservative for our datasets.

**Comparison with RS-HDP-HMM.** RS-HDP-HMM (Słupiński & Lipiński, 2025) represents an alternative approach to incorporating observation-dependence: it makes the self-persistence probability (the $\kappa$ parameter in sticky HDP-HMM terminology) depend on observations via logistic regression, while retaining implicit geometric durations. In contrast, RED-HDP-HMM models observation-dependent *durations* explicitly through negative binomial regression. Our results suggest that explicit duration modeling provides additional benefits: RED-HDP-HMM outperforms RS-HDP-HMM on all 7 benchmarks, with the largest gains on real-world data where duration distributions are likely non-geometric. The complementary nature of these approaches suggests that combining recurrent transitions with explicit durations could yield further improvements.

## 7.1. NASCAR

The NASCAR benchmark (Linderman et al., 2017) generates oval trajectories controlled by four discrete states (Figure 2). DS-HDP-HMM achieves the highest accu-

*Table 2.* Segmentation results (mean ± std over 5 independent MCMC runs). Best results in **bold**.

| Model | Accuracy | Weighted F1 |
|---|---|---|
| | Bee Dance | |
| S-HDP-HMM | $0.873 \pm 0.030$ | $0.856 \pm 0.043$ |
| DS-HDP-HMM | $0.842 \pm 0.070$ | $0.825 \pm 0.072$ |
| RS-HDP-HMM | $0.853 \pm 0.971$ | $0.871 \pm 0.872$ |
| RED-HDP-HMM (Nonloopy) | $\mathbf{0.899 \pm 0.028}$ | $\mathbf{0.884 \pm 0.042}$ |
| RED-HDP-HMM (Loopy) | $0.877 \pm 0.022$ | $0.862 \pm 0.030$ |
| HDP-HSMM (Poisson) | $0.327 \pm 0.025$ | $0.223 \pm 0.016$ |
| HDP-HSMM (NB) | $0.335 \pm 0.013$ | $0.244 \pm 0.012$ |
| | Synthetic NASCAR | |
| S-HDP-HMM | $0.942 \pm 0.062$ | $0.953 \pm 0.040$ |
| DS-HDP-HMM | $\mathbf{0.982 \pm 0.009}$ | $\mathbf{0.981 \pm 0.009}$ |
| RS-HDP-HMM | $0.921 \pm 0.059$ | $0.931 \pm 0.046$ |
| RED-HDP-HMM (Nonloopy) | $0.918 \pm 0.074$ | $0.927 \pm 0.063$ |
| RED-HDP-HMM (Loopy) | $0.943 \pm 0.059$ | $0.953 \pm 0.038$ |
| HDP-HSMM (Poisson) | $0.329 \pm 0.011$ | $0.253 \pm 0.012$ |
| HDP-HSMM (NB) | $0.325 \pm 0.015$ | $0.252 \pm 0.018$ |
| | Neural Data (0) | |
| S-HDP-HMM | $0.422 \pm 0.022$ | $0.330 \pm 0.027$ |
| DS-HDP-HMM | $0.408 \pm 0.020$ | $0.299 \pm 0.022$ |
| RS-HDP-HMM | $0.388 \pm 0.020$ | $0.277 \pm 0.021$ |
| RED-HDP-HMM (Nonloopy) | $\mathbf{0.454 \pm 0.012}$ | $\mathbf{0.366 \pm 0.015}$ |
| RED-HDP-HMM (Loopy) | $0.428 \pm 0.012$ | $0.328 \pm 0.011$ |
| HDP-HSMM (Poisson) | $0.366 \pm 0.005$ | $0.317 \pm 0.008$ |
| HDP-HSMM (NB) | $0.358 \pm 0.021$ | $0.309 \pm 0.019$ |
| | Neural Data (1) | |
| S-HDP-HMM | $0.351 \pm 0.015$ | $0.239 \pm 0.008$ |
| DS-HDP-HMM | $0.334 \pm 0.020$ | $0.216 \pm 0.021$ |
| RS-HDP-HMM | $0.343 \pm 0.028$ | $0.224 \pm 0.031$ |
| RED-HDP-HMM (Nonloopy) | $\mathbf{0.423 \pm 0.015}$ | $0.320 \pm 0.022$ |
| RED-HDP-HMM (Loopy) | $0.382 \pm 0.020$ | $0.281 \pm 0.027$ |
| HDP-HSMM (Poisson) | $0.362 \pm 0.026$ | $\mathbf{0.322 \pm 0.028}$ |
| HDP-HSMM (NB) | $0.338 \pm 0.015$ | $0.297 \pm 0.011$ |
| | Neural Data (2) | |
| S-HDP-HMM | $0.421 \pm 0.018$ | $0.332 \pm 0.020$ |
| DS-HDP-HMM | $0.379 \pm 0.013$ | $0.292 \pm 0.014$ |
| RS-HDP-HMM | $0.402 \pm 0.022$ | $0.317 \pm 0.028$ |
| RED-HDP-HMM (Nonloopy) | $\mathbf{0.456 \pm 0.042}$ | $\mathbf{0.399 \pm 0.047}$ |
| RED-HDP-HMM (Loopy) | $0.442 \pm 0.027$ | $0.374 \pm 0.030$ |
| HDP-HSMM (Poisson) | $0.367 \pm 0.014$ | $0.327 \pm 0.015$ |
| HDP-HSMM (NB) | $0.367 \pm 0.009$ | $0.324 \pm 0.008$ |
| | Neural Data (3) | |
| S-HDP-HMM | $0.395 \pm 0.033$ | $0.291 \pm 0.043$ |
| DS-HDP-HMM | $0.385 \pm 0.034$ | $0.267 \pm 0.033$ |
| RS-HDP-HMM | $0.384 \pm 0.013$ | $0.268 \pm 0.014$ |
| RED-HDP-HMM (Nonloopy) | $\mathbf{0.477 \pm 0.039}$ | $\mathbf{0.373 \pm 0.048}$ |
| RED-HDP-HMM (Loopy) | $0.419 \pm 0.037$ | $0.311 \pm 0.035$ |
| HDP-HSMM (Poisson) | $0.390 \pm 0.036$ | $0.311 \pm 0.029$ |
| HDP-HSMM (NB) | $0.376 \pm 0.013$ | $0.303 \pm 0.007$ |
| | Neural Data (4) | |
| S-HDP-HMM | $0.422 \pm 0.013$ | $0.355 \pm 0.024$ |
| DS-HDP-HMM | $0.380 \pm 0.017$ | $0.302 \pm 0.017$ |
| RS-HDP-HMM | $0.398 \pm 0.019$ | $0.316 \pm 0.020$ |
| RED-HDP-HMM (Nonloopy) | $\mathbf{0.485 \pm 0.013}$ | $\mathbf{0.452 \pm 0.016}$ |
| RED-HDP-HMM (Loopy) | $0.431 \pm 0.025$ | $0.369 \pm 0.033$ |
| HDP-HSMM (Poisson) | $0.358 \pm 0.013$ | $0.296 \pm 0.014$ |
| HDP-HSMM (NB) | $0.348 \pm 0.015$ | $0.289 \pm 0.014$ |

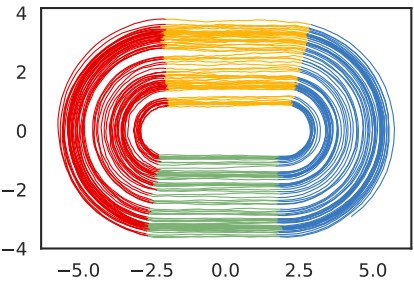

*Figure 2.* NASCAR benchmark trajectory with four latent states controlling 2D dynamics.

where the disentangled sticky mechanism alone suffices.

### 7.2. Bee Dance

The Bee Dance dataset (Oh et al., 2008) captures honey-bee waggle dances, where bees communicate food source locations through sequences of waggle, left-turn, and right-turn movements. Observations include 2D coordinates and head orientation: $\boldsymbol{y}_t = [\cos(\theta_t), \sin(\theta_t), x_t, y_t]^T$. Dance segment durations correlate with food source distance, violating geometric duration assumptions.

RED-HDP-HMM (Nonloopy) achieves 0.899 accuracy, a 2.6 percentage point improvement over S-HDP-HMM (0.873). Notably, RED-HDP-HMM also outperforms RS-HDP-HMM (0.853), demonstrating that explicit duration modeling provides benefits beyond recurrent transition dynamics alone. The explicit duration modeling captures the relationship between dance movements and their characteristic lengths.

### 7.3. Neural Data

The C. elegans neural dataset (Kato et al., 2015) records calcium imaging from head ganglia neurons. An expert labeled 8 behavioral states (forward, reverse, turn, etc.). We apply PCA to reduce to 5 dimensions and fit separate models per recording. This dataset is well-suited for recurrent models due to the relationship between neural activity patterns and behavioral state duration (Linderman et al., 2019; Yezerets et al., 2024).

RED-HDP-HMM achieves the largest improvements on neural data: 3–8 percentage point gains over baselines (Table 2). The Neural Data (4) result (0.485 vs. 0.422 for S-HDP-HMM) represents a 15% relative improvement. Across all five neural recordings, RED-HDP-HMM consistently outperforms both sticky variants and RS-HDP-HMM, suggesting that explicit observation-dependent durations capture temporal structure that recurrent transitions alone cannot. These gains confirm that recurrent connections benefit neural activity modeling, extending prior findings (Linderman et al., 2019; Zoltowski et al., 2020) to the Bayesian nonpara-

racy (0.982), with RED-HDP-HMM (Loopy) competitive at 0.943. The explicit duration mechanism provides minimal benefit on this synthetic data with regular state transitions,

metric setting.

### 7.4. Summary

RED-HDP-HMM (Nonloopy) achieves the best performance on 6 of 7 benchmarks (with DS-HDP-HMM winning on NASCAR, where regular state transitions favor simpler models), with consistent improvements in both accuracy and weighted F1. Importantly, RED-HDP-HMM outperforms RS-HDP-HMM across all benchmarks, demonstrating that explicit duration modeling provides benefits beyond recurrent transition dynamics. The gains are most pronounced on real-world data where observation-duration relationships exist. These results support our theoretical contribution that RED-HDP-HMM strictly extends HDP-HSMM in expressiveness.

## 8. Limitations and Practical Implications

**Identifiability Considerations.**   A fundamental challenge in switching state-space models is identifiability: distinct parameter configurations may produce identical observation distributions. As demonstrated by Fox (2009) for switching vector autoregressive (VAR) processes - the model class to which RED-HDP-HMM belongs - even when restricting the number of modes $K$ or autoregressive order, the identification problem remains challenging. Multiple distinct parameterizations can generate identical observation sequences.

This non-identifiability manifests in several forms for RED-HDP-HMM:

- **Label permutation**: Any relabeling of states yields an equivalent model. This is inherent to all mixture and hidden state models and is handled through standard post-hoc alignment techniques such as Hungarian matching.

- **Duration-transition aliasing**: In loopy variants where self-transitions are permitted, extended sojourns can arise either from long sampled durations or from sequences of short durations with self-transitions. The nonloopy variant we emphasize avoids this aliasing by construction.

- **Emission-duration coupling**: When observations influence both emissions (via AR dynamics) and durations (via regression), certain symmetric parameter configurations may yield equivalent distributions.

Following the Bayesian nonparametric perspective of Fox (2009), we address identifiability through prior specification rather than hard constraints. The hierarchical Dirichlet process prior penalizes model complexity, providing automatic regularization that favors parsimonious explanations among observationally equivalent models.

In practice, we separate theoretical non-identifiability from segmentation instability. The five independent runs in Table 2 are summarized after label alignment, so benign label permutations do not affect the reported accuracy or weighted F1. Large run-to-run variance would indicate unstable segmentations rather than merely relabeled states; the reported standard deviations therefore provide a coarse empirical check that the posterior summaries are not driven by a single favorable run. This does not prove identifiability, and state-specific duration coefficients should still be interpreted only up to the usual label-alignment ambiguity.

Establishing formal identifiability results for observation-dependent duration models remains an important direction for future work.

**Practical Guidance.**   The *truncation error* (Theorem 5.1) is negligible in practice: with $L = 50$ and typical hyperparameters, the error is $\approx 3 \times 10^{-7}$. The expressiveness result (Theorem 5.4) applies to NB durations; for other duration families, RED-HDP-HMM still provides observation-dependence but formal separation guarantees would require additional analysis. The computational cost $\mathcal{O}(TL(L + D_{\max}))$ scales linearly with maximum duration; for applications with very long segments, practitioners should choose $D_{\max}$ based on domain knowledge.

## 9. Conclusions

We have introduced RED-HDP-HMM, the first Bayesian nonparametric hidden Markov model that unifies three capabilities previously available only in isolation: infinite state space modeling, explicit non-geometric duration distributions, and observation-dependent duration dynamics. By conditioning state durations on past observations through Gamma-mixed Negative Binomial regression, our model captures temporal dependencies that existing HDP-HMM variants fundamentally cannot represent - a relationship we quantify through strict expressiveness guarantees.

**Theoretical Foundations.**   Our analysis establishes that RED-HDP-HMM is a principled model with formal guarantees:

- **Truncation Error** (Theorem 5.1): The weak-limit approximation incurs exponentially vanishing error $O(\exp(-c'L))$, with explicit constants showing $L = 50$ yields error below $10^{-6}$.

- **Expressiveness** (Theorem 5.4): RED-HDP-HMM strictly contains HDP-HSMM, with strictly positive TV separation from any HDP-HSMM for models with observation-dependent durations.

Additionally, the Supplementary Material proves correctness of the forward-backward sampling algorithm (Theorem 4).

**Algorithmic and Empirical Contributions.** Our weak-limit Gibbs sampler achieves $\mathcal{O}(TL(L + D_{\max}))$ complexity by exploiting the deterministic countdown structure of duration variables. Experiments across synthetic data (NASCAR), behavioral data (Bee Dance), and neural recordings (C. elegans) demonstrate consistent improvements, with accuracy gains of 3–8 percentage points on neural data over sticky, disentangled sticky, and recurrent sticky HDP-HMM baselines.

Our experiments also validate two findings of independent interest: (i) nonloopy transitions yield superior segmentation when combined with infinite-support duration distributions, confirming prior theoretical insights; and (ii) recurrent observation-dependence benefits extend to the Bayesian nonparametric setting, confirming hypotheses from finite-state models (Linderman et al., 2019; Zoltowski et al., 2020).

**Limitations.** As discussed in Section 8, identifiability in switching models is a fundamental challenge that we address through Bayesian regularization rather than hard constraints. Computational cost scales linearly with $D_{\max}$, requiring practitioners to set reasonable duration bounds for applications with very long segments.

**Future Directions.** Natural extensions include integrating RED-HDP-HMM with switching linear dynamical systems for joint discrete-continuous latent inference - paralleling the rSLDS (Linderman et al., 2017) and REDSLDS (Słupiński & Lipiński, 2024a) lines of work, but lifting the finite-state constraint. On the emission side, VAR($p$) models are a direct conjugate extension of the AR(1) specification used here, while nonlinear or neural emissions would trade conjugacy for expressiveness. Developing variational inference would bring scalability to massive time series, bridging the gap between Bayesian nonparametric rigor and the computational efficiency achieved by deep switching models such as SNLDS (Dong et al., 2020) and REDSDS (Ansari et al., 2021) through amortized inference. Two open problems remain particularly pressing: establishing formal identifiability conditions for observation-dependent duration models, and developing scalable BNP inference for semi-Markov models - a setting that remains underexplored relative to its parametric counterpart. RED-HDP-HMM demonstrates that observation-dependent duration modeling, long recognized as important in finite-state settings, provides substantial benefits in the Bayesian nonparametric framework.

## Acknowledgements

We gratefully acknowledge Polish high-performance computing infrastructure PLGrid (HPC Center: ACK Cyfronet AGH) for providing computer facilities and support within computational grant no. PLG/2024/017620.

This work was supported by the Polish National Science Centre (NCN) under grant OPUS-18 no. 2019/35/B/ST6/04379.

## Impact Statement

This paper presents work whose goal is to advance the field of Machine Learning. There are many potential societal consequences of our work, none which we feel must be specifically highlighted here.

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

**Theorem .2** (Truncation Error for Finite Horizons). *Let $\Pi_\infty^{(T)}$ denote the marginal distribution over observation sequences of length $T$ under the full infinite-state RED-HDP-HMM, integrating out the transition parameters $\{\pi_j\}$. Let $\Pi_L^{(T)}$ denote the distribution under the aggregated L-lumped model.*

*Assume the initial state $z_1$ in the full model is drawn from the marginal distribution induced by the global weights $\boldsymbol{\beta}$ (i.e., $\mathbb{P}(z_1 = k \mid \boldsymbol{\beta}) = \beta_k$).*

*Define $c := \log((\gamma + 1)/\gamma)$. Then for all $L \geq 2$:*

$$d_{\mathrm{TV}}\left(\Pi_L^{(T)}, \Pi_\infty^{(T)}\right) \leq T\, e^{-c(L-1)}.$$

*Proof.* We bound the total variation distance by constructing a coupling between the full process $(z_{1:T}, y_{1:T})$ and the lumped process $(z'_{1:T}, y'_{1:T})$. By the definition of total variation distance, it suffices to bound the probability that the observation sequences differ: $d_{\mathrm{TV}} \leq \mathbb{P}(y_{1:T} \neq y'_{1:T})$.

**Step 1: The Coupling Construction.** We couple the generative processes as follows:

1. **Parameters:** Sample global weights $\boldsymbol{\beta}$ via the stick-breaking process. For the full model, sample transition vectors $\pi_j \sim \mathrm{DP}(\alpha, \boldsymbol{\beta})$. Construct the lumped parameters $\pi^{(L)}$ deterministically according to Definition 1.

2. **Latent Dynamics:**

   Initialize $z_1 = z'_1$ from $\boldsymbol{\beta}$. Note that if $z_1 \geq L$, the processes diverge immediately.

   For $t > 1$:

   - If $z_{t-1} = z'_{t-1} = i$ where $i < L$:
     Sample a uniform random variable $u_t \sim \mathrm{Unif}[0, 1]$. Use $u_t$ to select the next state for both chains via the inverse CDF method.
   - Since $\pi_{ik} = \pi_{ik}^{(L)}$ for all $k < L$, and the mass of the tail in the full model ($\sum_{m=L}^\infty \pi_{im}$) equals the mass of state $L$ in the lumped model ($\pi_{iL}^{(L)}$), the states remain identical ($z_t = z'_t$) provided the full model does not enter the tail (any state $k \geq L$).
   - If the full model selects $k \geq L$, we set $z'_t = L$. From this point onward, $z'_t$ remains at $L$ (absorbing property), while $z_t$ continues to evolve in the full space. The coupling on states is lost.

3. **Emissions:** Let $F_L$ be an arbitrary emission distribution associated with the lumped state $L$. For every time step $t$:

   - If $z_t = z'_t < L$: We generate the observations $y_t$ and $y'_t$ using the *same* random seed and emission parameters $\theta_{z_t}$. Consequently, $z_t = z'_t \implies y_t = y'_t$.
   - If $z'_t = L$: We generate $y'_t \sim F_L$ and $y_t \sim F_{z_t}$ independently. In this case, we conservatively assume $y_t \neq y'_t$.

4. **Coupling Inequality:** Let $E$ be the event that the full process enters the tail (i.e., $E = \{\exists t \in \{1, \ldots, T\} : z_t \geq L\}$).

   By construction, if the full process never enters the tail (event $E^c$ occurs), then $z_t = z'_t$ for all $t$, which implies $y_{1:T} = y'_{1:T}$. If $E$ occurs, the two observation sequences may still coincide by chance, but we conservatively count every tail visit as a coupling failure. Thus $\{y_{1:T} \neq y'_{1:T}\} \subseteq E$, giving the second inequality below.

   Using the Coupling Lemma, which states that the total variation distance between two distributions is bounded by the probability that their coupled realizations differ, we have:

   $$d_{\text{TV}}\left(\Pi_L^{(T)}, \Pi_\infty^{(T)}\right) \ \leq \ \mathbb{P}(y_{1:T} \neq y'_{1:T}) \ \leq \ \mathbb{P}(E).$$

**Step 2: Expected Tail Mass.** Let $\tau_{L-1} := \sum_{k=L}^{\infty} \beta_k$ be the aggregate mass of the tail states. Using the stick-breaking construction $\beta_k = v_k \prod_{\ell=1}^{k-1}(1 - v_\ell)$ with $v_k \sim \text{Beta}(1, \gamma)$, the remaining stick length after $L - 1$ breaks is:

$$\tau_{L-1} = \prod_{\ell=1}^{L-1}(1 - v_\ell).$$

The expected tail mass is:

$$\mathbb{E}[\tau_{L-1}] = \prod_{\ell=1}^{L-1} \mathbb{E}[1 - v_\ell] = \prod_{\ell=1}^{L-1} \frac{\gamma}{\gamma + 1} = \left(\frac{\gamma}{\gamma + 1}\right)^{L-1} = e^{-c(L-1)}.$$

**Step 3: Marginal Stationarity and Union Bound.** We must bound $\mathbb{P}(z_t \geq L)$. In the HDP-HMM, transitions are governed by $\pi_j \sim \text{DP}(\alpha, \boldsymbol{\beta})$. A key property of the Dirichlet Process is that the expected transition vector equals the base measure:

$$\mathbb{E}[\pi_{jk} \mid \boldsymbol{\beta}] = \beta_k.$$

We prove by induction that if the marginal distribution of $z_{t-1}$ is $\boldsymbol{\beta}$, then the marginal distribution of $z_t$ is also $\boldsymbol{\beta}$.

$$\begin{aligned} P(z_t = k \mid \boldsymbol{\beta}) &= \sum_{j=1}^{\infty} P(z_t = k \mid z_{t-1} = j, \boldsymbol{\beta}) P(z_{t-1} = j \mid \boldsymbol{\beta}) \\ &= \sum_{j=1}^{\infty} \mathbb{E}[\pi_{jk} \mid \boldsymbol{\beta}] \beta_j \\ &= \sum_{j=1}^{\infty} \beta_k \beta_j = \beta_k \left(\sum_{j=1}^{\infty} \beta_j\right) = \beta_k. \end{aligned}$$

Therefore, for any single time step $t$, the probability of entering the tail matches the expected tail mass of $\boldsymbol{\beta}$:

$$\mathbb{P}(z_t \geq L) = \mathbb{E}_{\boldsymbol{\beta}}[\mathbb{E}[z_t \geq L \mid \boldsymbol{\beta}]] = \mathbb{E}_{\boldsymbol{\beta}}\left[\sum_{k=L}^{\infty} \beta_k\right] = e^{-c(L-1)}.$$

Applying the union bound over time steps $t = 1, \ldots, T$:

$$\mathbb{P}(\exists t : z_t \geq L) \leq \sum_{t=1}^{T} \mathbb{P}(z_t \geq L) = T \, e^{-c(L-1)}.$$

$\square$

### .1. Numerical Validation of State Truncation Bound (Theorem 1)

Theorem 5.1 bounds the error from truncating the infinite HDP to $L$ states. The bound $d_{\mathrm{TV}} \leq T \cdot e^{-c(L-1)}$ with $c = \log((\gamma + 1)/\gamma)$ derives from the GEM$(\gamma)$ stick-breaking prior, where the tail mass $\tau_L = \sum_{k>L} \beta_k$ satisfies $\mathbb{E}[\tau_L] = (\gamma/(\gamma + 1))^L = e^{-cL}$, and the shift to $L - 1$ accounts for the lumped state requiring $L \geq 2$.

We validate this by simulating the stick-breaking process: $\beta_k = V_k \prod_{j<k}(1 - V_j)$ where $V_k \sim \mathrm{Beta}(1, \gamma)$, and measuring the empirical tail mass.

*Listing 1.* State truncation validation via GEM$(\gamma)$ stick-breaking simulation.

```python
import numpy as np
from scipy.stats import beta

def simulate_gem_tail_mass(gamma, L_max, num_simulations=10000):
    """Simulate GEM(gamma) and compute tail mass tau_L.
    Key insight: tau_L = prod_{k=1}^L (1-V_k) exactly.
    Uses log-space for numerical stability."""
    V = beta.rvs(1, gamma, size=(num_simulations, L_max))
    # log(tau_L) = sum_{k=1}^L log(1-V_k)
    log_one_minus_V = np.log(1 - V)
    log_tail_masses = np.cumsum(log_one_minus_V, axis=1)
    return np.exp(log_tail_masses)

# Theoretical bound: E[tau_L] = exp(-cL), c = log((gamma+1)/gamma)
def theoretical_bound(gamma, L):
    c = np.log((gamma + 1) / gamma)
    return np.exp(-c * L)

def compute_ratio_log_space(emp, theo, min_val=1e-300):
    """Compute ratio in log-space for numerical stability."""
    log_ratio = np.log(np.maximum(emp, min_val)) - np.log(np.maximum(theo, min_val))
    return np.exp(log_ratio)

# Run for gamma = 0.5, 1.0, 2.0, 4.0, 8.0, 16.0
for gamma in [0.5, 1.0, 2.0, 4.0, 8.0, 16.0]:
    tail_masses = simulate_gem_tail_mass(gamma, L_max=60)
    empirical = np.mean(tail_masses, axis=0)
    ratio = compute_ratio_log_space(empirical, theoretical_bound(gamma, L))
```

**Console Output.**

```
SUMMARY: Empirical vs Theoretical at L=50
----------------------------------------------------------------------
     gamma |        c  | Empirical E[tau_L] |   Theoretical  | Ratio
----------------------------------------------------------------------
       0.5 |  1.098612 |     6.19e-29       |   1.39e-24     | 0.00
       1.0 |  0.693147 |     1.93e-16       |   8.88e-16     | 0.22
       2.0 |  0.405465 |     1.44e-09       |   1.57e-09     | 0.92
       4.0 |  0.223144 |     1.41e-05       |   1.43e-05     | 0.99
       8.0 |  0.117783 |     2.75e-03       |   2.77e-03     | 0.99
      16.0 |  0.060625 |     4.86e-02       |   4.83e-02     | 1.01
----------------------------------------------------------------------
Ratio computed via log-space: exp(log(emp) - log(theo)) for stability
```

**Discussion.** The simulation confirms the exponential decay predicted by Theorem 5.1 across all tested concentration parameters $\gamma \in \{0.5, 1, 2, 4, 8, 16\}$.

*Decay rate verification.* The decay rate $c = \log((\gamma + 1)/\gamma)$ decreases with increasing $\gamma$: from $c \approx 1.10$ for $\gamma = 0.5$ down to $c \approx 0.06$ for $\gamma = 16$. For moderate values ($\gamma \in \{2, 4, 8, 16\}$), the empirical-to-theoretical ratio remains close to 1.0 (within 1–8%) across all truncation levels $L \leq 50$, validating our theoretical bound.

*Numerical precision.* For small $\gamma$ values ($\gamma \leq 1$), the rapid decay causes $\tau_L$ to reach machine epsilon around $L = 30$, after which the empirical estimates become unreliable (ratio near 0). We compute ratios in log-space $(\exp(\log(\mathrm{emp}) - \log(\mathrm{theo})))$ for numerical stability when both values are extremely small. This instability is a Monte Carlo artifact, not a failure of the bound.

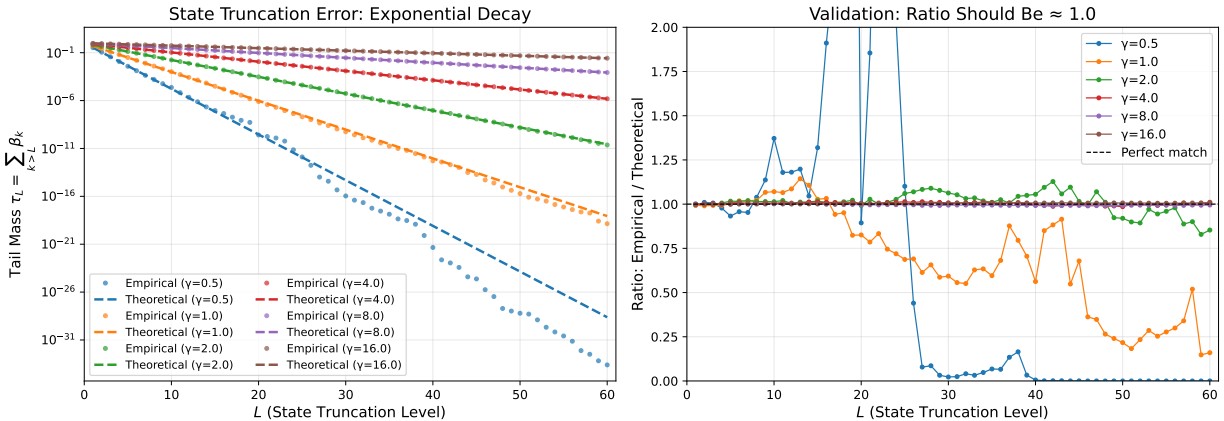

*Figure 3.* Numerical validation of Theorem 5.1. **Left:** Empirical tail mass $\tau_L$ (circles) versus theoretical bound $e^{-cL}$ (dashed lines) for six concentration parameters ($\gamma \in \{0.5, 1, 2, 4, 8, 16\}$). Both decay exponentially with rates $c = \log((\gamma + 1)/\gamma)$. **Right:** Ratio of empirical to theoretical values (y-axis: 0–2), computed in log-space for numerical stability. For $\gamma \geq 2$, the ratio remains near 1.0 across all $L$. For smaller $\gamma$, the faster decay ($c \approx 1.1$ for $\gamma = 0.5$) causes the empirical estimator variance to dominate at large $L$. Larger $\gamma$ values yield slower decay but maintain excellent agreement with theory throughout.

*Large $\gamma$ regime.* For $\gamma \geq 4$, the slower decay ($c \leq 0.22$) means larger truncation levels are needed for comparable error bounds. With $\gamma = 16$, $\tau_{50} \approx 0.05$—requiring $L \approx 150$ to achieve $\tau_L < 10^{-4}$. This tradeoff is important when the prior expects many active states.

*Practical implication.* For typical choices ($\gamma \in [1, 4]$), truncation to $L = 50$ yields truncation error $\tau_L \in [10^{-16}, 10^{-5}]$—negligible for realistic sequence lengths. When using larger $\gamma$ (e.g., $\gamma = 16$), practitioners should increase $L$ proportionally to maintain the same error tolerance.

## A. Theorem 2: Duration Truncation Error Bound

**Theorem A.1** (Duration Truncation Error Conditional on Parameters). *Let $\Theta$ be the set of parameters including duration parameters $\{r_j, p_j(\cdot)\}_{j=1}^{\infty}$. Assume there exists $\epsilon > 0$ such that for all states $j$ and contexts $y$, the success probability satisfies $p_j(y) \in [\epsilon, 1 - \epsilon]$. **Assume dispersion parameters are uniformly bounded:** there exists $R < \infty$ such that $r_j \leq R$ for all $j$.*

*Let $\Pi^{(T)}(\cdot \mid \Theta)$ be the probability measure on trajectories of length $T$ under the full semi-Markov model. Let $\Pi_{D_{\max}}^{(T)}(\cdot \mid \Theta)$ be the measure under the clamped model, where the duration distribution is defined by the censored random variable $d' = \min(d, D_{\max})$.*

*Then there exist constants $C(R, \epsilon) < \infty$ and $\lambda(\epsilon) > 0$ such that the Total Variation distance satisfies:*

$$d_{\text{TV}}\left(\Pi_{D_{\max}}^{(T)}(\cdot \mid \Theta), \Pi^{(T)}(\cdot \mid \Theta)\right) \leq C(R, \epsilon)\, T\, e^{-\lambda(\epsilon) D_{\max}}.$$

*Proof.* We establish the bound by constructing a maximal coupling between the full process $(Y_{full})$ and the clamped process $(Y_{clamped})$.

**Step 1: Uniform Exponential Tail Bound.** We first establish a tail bound uniform across all states and contexts. Let $d \sim \text{NB}(r, p) + 1$, representing the number of failures before $r$ successes, shifted by 1. The Moment Generating Function (MGF) of the unshifted Negative Binomial component is $M_{NB}(t) = \left(\frac{p}{1-(1-p)e^t}\right)^r$. The MGF for $d$ is therefore:

$$M_d(t) = \mathbb{E}[e^{t(NB+1)}] = e^t \left(\frac{p}{1 - (1-p)e^t}\right)^r.$$

This MGF converges provided that $(1 - p)e^t < 1$, or equivalently $t < -\log(1 - p)$. Given the assumption $p \geq \epsilon$, we have $-\log(1 - p) \geq -\log(1 - \epsilon)$. We choose the rate parameter:

$$\lambda(\epsilon) = -\frac{1}{2}\log(1 - \epsilon).$$

This choice ensures $\lambda(\epsilon)$ is strictly inside the region of convergence. We define the uniform bound constant $C(R, \epsilon)$ as:

$$C(R, \epsilon) = \sup_{\substack{r \leq R \\ p \in [\epsilon, 1-\epsilon]}} M_{\mathrm{NB}(r,p)+1}(\lambda(\epsilon)).$$

Since the parameters lie in a compact subset and the MGF is continuous within the convergence radius, $C(R, \epsilon) < \infty$. By the Chernoff bound, for any state $j$ and context $y$:

$$\mathbb{P}(d_{j,y} > D_{\max}) \leq e^{-\lambda(\epsilon)D_{\max}} M_{d_{j,y}}(\lambda(\epsilon)) \leq C(R, \epsilon)e^{-\lambda(\epsilon)D_{\max}}. \tag{13}$$

**Step 2: Construction of the Coupling.** To handle the random number of transitions rigorously, we construct the coupling using an infinite sequence of potential draws. Let $\omega = \{u_k\}_{k=1}^{\infty}$ be a sequence of i.i.d. random variables (seeds) used to generate durations. For any index $k$, state $s$, and context $y$, let $d(u_k, s, y)$ be a draw from $\mathrm{NB}(r_s, p_s(y)) + 1$.

We define the coupled processes $Y_{full}$ and $Y_{clamped}$ as follows:

1. Both processes initialize at the same state $s_1$ at time $t = 0$.

2. For the $k$-th transition from state $s_k$ with context $y_k$:
   - We generate the "raw" duration $d_k = d(u_k, s_k, y_k)$.
   - $Y_{full}$ sets duration $\tau_k = d_k$.
   - $Y_{clamped}$ sets duration $\tau_k' = \min(d_k, D_{\max})$.

3. If $\tau_k = \tau_k'$ (i.e., $d_k \leq D_{\max}$), the processes remain coupled: clocks advance by the same amount, and the next state $s_{k+1}$ is generated identically.

4. If $\tau_k \neq \tau_k'$ (i.e., $d_k > D_{\max}$), the coupling breaks.

**Step 3: Bounding the Coupling Failure.** By the coupling inequality, the Total Variation distance is bounded by the probability that the realized trajectories differ within the horizon $T$:

$$d_{\mathrm{TV}} \leq \mathbb{P}(Y_{full} \neq Y_{clamped}).$$

Let $N$ be the random number of segments occurring in the full process up to time $T$. The processes differ if and only if a duration is clamped at some step $k \leq N$. Thus:

$$\{Y_{full} \neq Y_{clamped}\} = \bigcup_{k=1}^{N} \{d_k > D_{\max}\}.$$

Since every duration satisfies $d_k \geq 1$, the number of transitions is bounded almost surely by the time horizon: $N \leq T$. Therefore, we have the set inclusion:

$$\bigcup_{k=1}^{N} \{d_k > D_{\max}\} \subseteq \bigcup_{k=1}^{T} \{d_k > D_{\max}\}.$$

This allows us to work with a fixed number of terms $T$ rather than a random index $N$.

**Step 4: Conclusion.**    We apply the union bound to the fixed set of indices:

$$\mathbb{P}\left(\bigcup_{k=1}^{T}\{d_k > D_{\max}\}\right) \leq \sum_{k=1}^{T}\mathbb{P}(d_k > D_{\max}).$$

We evaluate each term $\mathbb{P}(d_k > D_{\max})$ by conditioning on the history $\mathcal{H}_{k-1}$ (which determines the state $s_k$ and context $y_k$ at step $k$). By the Law of Total Probability:

$$\mathbb{P}(d_k > D_{\max}) = \mathbb{E}_{\mathcal{H}_{k-1}}\left[\mathbb{P}(d_k > D_{\max} \mid s_k, y_k)\right].$$

Using the uniform bound established in Eq. 13, we know that for *any* state $s$ and context $y$:

$$\mathbb{P}(d_k > D_{\max} \mid s, y) \leq C(R, \epsilon)e^{-\lambda(\epsilon)D_{\max}}.$$

Since this bound is uniform, it passes through the expectation:

$$\mathbb{P}(d_k > D_{\max}) \leq \mathbb{E}_{\mathcal{H}_{k-1}}\left[C(R, \epsilon)e^{-\lambda(\epsilon)D_{\max}}\right] = C(R, \epsilon)e^{-\lambda(\epsilon)D_{\max}}.$$

Summing over $k = 1$ to $T$:

$$d_{\mathrm{TV}} \leq \sum_{k=1}^{T} C(R, \epsilon)\, e^{-\lambda(\epsilon)D_{\max}} = T\, C(R, \epsilon)\, e^{-\lambda(\epsilon)D_{\max}}.$$

This completes the proof. $\qquad\square$

**Corollary (Joint State-Duration Truncation).** *For the* $(L, D_{\max})$-*truncated RED-HDP-HMM:*

$$d_{\mathrm{TV}}\left(\Pi_{L,D_{\max}}^{(T)}, \Pi_{\infty}^{(T)}\right) \;\leq\; T\, e^{-c(L-1)} + C(R, \epsilon)\, T\, e^{-\lambda(\epsilon)D_{\max}}.$$

*Proof.* By the triangle inequality:

$$d_{\mathrm{TV}}\left(\Pi_{L,D_{\max}}^{(T)}, \Pi_{\infty}^{(T)}\right) \;\leq\; d_{\mathrm{TV}}\left(\Pi_{L,D_{\max}}^{(T)}, \Pi_{L}^{(T)}\right) + d_{\mathrm{TV}}\left(\Pi_{L}^{(T)}, \Pi_{\infty}^{(T)}\right).$$

The first term is the duration truncation error for the $L$-truncated model, bounded by Theorem 5.2. The second term is the state truncation error, bounded by Theorem 5.1. The result follows by adding these bounds. $\qquad\square$

### A.1. Numerical Validation of Duration Truncation Bound

To empirically validate Theorem A.1, we implement a coupling-based simulation that directly measures the probability of divergence between the full (unclamped) and truncated (clamped) semi-Markov processes. The key insight is that under a maximal coupling, both processes share identical random seeds for duration generation; they diverge only when a raw duration exceeds $D_{\max}$, at which point the clamped process truncates while the full process does not. The probability of such divergence within horizon $T$ is exactly the quantity bounded by our theorem.

*Listing 2.* Coupling-based validation of the duration truncation bound. The simulation generates 5000 coupled trajectory pairs for each $D_{\max}$ value and measures the empirical divergence probability.

```
1   import matplotlib
2   matplotlib.use('Agg')
3   import numpy as np
4   import matplotlib.pyplot as plt
5   from scipy.stats import nbinom
6
7   class SemiMarkovSimulation:
8       def __init__(self, r_params, p_params, epsilon, R_bound):
9           self.r = r_params
10          self.p = p_params
11          self.epsilon = epsilon
12          self.lam = -0.5 * np.log(1 - self.epsilon)
13
14          # Calculate C(R, epsilon) via MGF supremum (log-space for stability)
15          log_mgf_values = []
```

```
16            for r_val, p_val in zip(self.r, self.p):
17                term_denom = 1 - (1 - p_val) * np.exp(self.lam)
18                log_m_nb = r_val * (np.log(p_val) - np.log(term_denom))
19                log_m_d = self.lam + log_m_nb
20                log_mgf_values.append(log_m_d)
21            self.log_C = max(log_mgf_values)
22            self.C_constant = np.exp(self.log_C)
23
24        def generate_trajectory(self, T, D_max, seed):
25            rng = np.random.default_rng(seed)
26            current_time, current_state = 0.0, 0
27            while current_time < T:
28                r_val, p_val = self.r[current_state], self.p[current_state]
29                raw_d = nbinom.rvs(n=r_val, p=p_val, random_state=rng) + 1
30                if raw_d > D_max:
31                    return True  # Diverged
32                current_time += raw_d
33                current_state = 1 - current_state
34            return False
35
36        def run_experiment(self, T, d_max_range, num_simulations=5000):
37            empirical_tv, theoretical_bound = [], []
38            for D in d_max_range:
39                failures = sum(1 for i in range(num_simulations)
40                            if self.generate_trajectory(T, D, seed=i))
41                empirical_tv.append(failures / num_simulations)
42                # Compute bound in log-space for stability
43                log_bound = self.log_C + np.log(T) - self.lam * D
44                theoretical_bound.append(np.exp(log_bound))
45            return empirical_tv, theoretical_bound
46
47    # Configuration: two-state semi-Markov
48    sim = SemiMarkovSimulation(r_params=[2, 4], p_params=[0.3, 0.7],
49                            epsilon=0.2, R_bound=5)
50    empirical, theoretical = sim.run_experiment(T=20, d_max_range=range(1, 25))
```

**Console Output.**    Running the simulation produces the following diagnostic output and results:

```
==============================================================
SIMULATION PARAMETERS
==============================================================
States: 2
  State 0: NB(r=2, p=0.3)
  State 1: NB(r=4, p=0.7)
epsilon = 0.2
R_bound = 5

THEORETICAL CONSTANTS
------------------------------------------
lambda(epsilon) = -0.5 * log(1 - 0.2)
              = 0.111572
C(R, epsilon)  = max MGF at lambda
              = 2.129480
==============================================================

Running 5000 simulations for each D_max value...
Time horizon T = 20

RESULTS
==============================================================
 D_max |   Empirical TV |  Theoretical Bound |   Ratio
--------------------------------------------------------------
     1 |       1.000000 |          38.093291 |   0.0263
     5 |       0.909400 |          24.379706 |   0.0373
    10 |       0.324800 |          13.955758 |   0.0233
```

```
15 |        0.074200  |          7.988742  |    0.0093
20 |        0.017000  |          4.573023  |    0.0037
24 |        0.004800  |          2.926735  |    0.0016
========================================================================
```

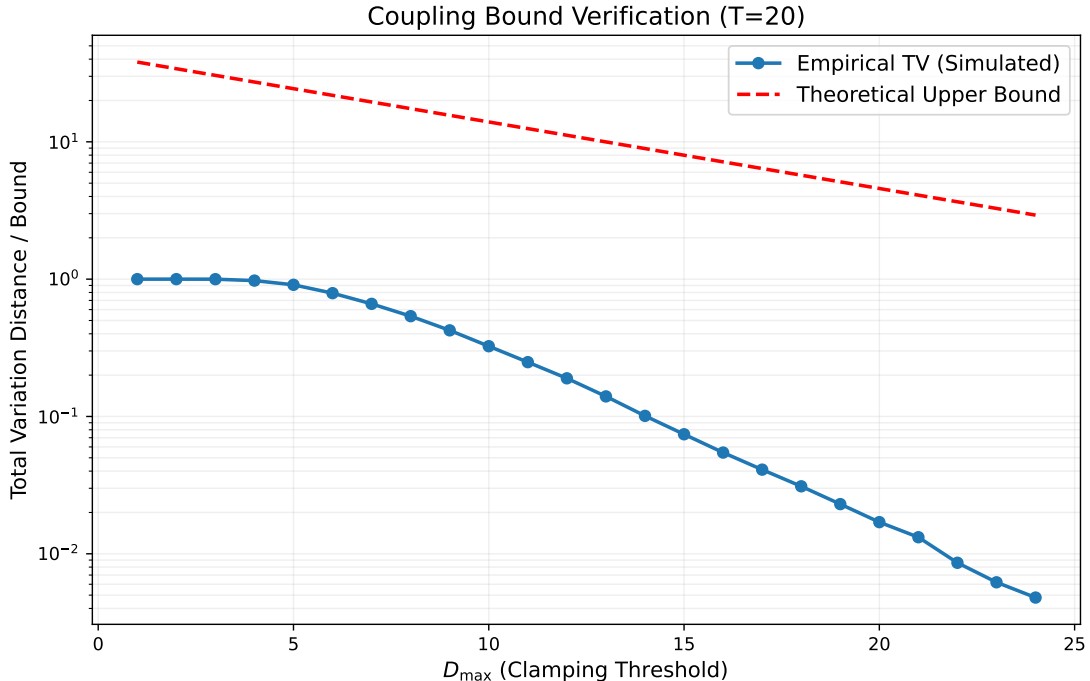

*Figure 4.* Numerical validation of Theorem A.1. The empirical TV distance (blue circles) represents the fraction of 5000 coupled trajectory pairs that diverged within horizon $T = 20$. The theoretical upper bound (red dashed line) is $C(R, \epsilon) \cdot T \cdot e^{-\lambda(\epsilon)D_{\max}}$ with $\lambda \approx 0.112$ and $C \approx 2.13$. Both curves decay exponentially (note log scale), with the empirical values consistently 1–2 orders of magnitude below the bound.

**Discussion.** The simulation validates Theorem A.1 across all tested $D_{\max}$ values. Several observations merit discussion:

*Bound validity.* The empirical TV distance (probability of coupling failure) lies strictly below the theoretical bound for all $D_{\max} \geq 1$, confirming the theorem's correctness. Both quantities exhibit the predicted exponential decay with rate $\lambda(\epsilon) \approx 0.112$.

*Bound conservatism.* The theoretical bound exceeds the empirical TV by a factor of 25–60×. This gap arises from the union bound over $T$ potential truncation events (Step 4 of the proof), which assumes independence of failure events across segments. In practice, (i) trajectories contain far fewer than $T$ segments since each duration $d \geq 1$, and (ii) early truncation events prevent later ones from contributing, making the union bound loose.

*Practical implications.* For this two-state example with $T = 20$ and moderate dispersion parameters, $D_{\max} \approx 15$ achieves empirical TV $< 0.08$, while the conservative bound gives TV $< 8$. The paper's recommended $D_{\max} = 200$ (Section 6) provides empirical TV on the order of $10^{-9}$ - effectively exact for practical purposes.

## B. Theorem 3: Strict Expressiveness over HDP-HSMM

**Theorem B.1** (Expressiveness with NB Durations). *The RED-HDP-HMM model class strictly contains the HDP-HSMM model class when both use Negative Binomial (NB) duration distributions. Specifically:*

1. ***Containment****: For any HDP-HSMM with NB duration parameters $(\boldsymbol{\beta}, \{\pi_j\}, \{\theta_j\}, \{(r_j, p_j)\})$, there exists a RED-HDP-HMM with the same joint distribution over states and observations.*

2. **Strictness**: *There exist distributions representable by RED-HDP-HMM with NB durations that cannot be represented by any HDP-HSMM with NB durations.*

*Proof.* **Part 1: Containment** Both models utilize the same semi-Markov generative framework: at each segment transition, a new state $z$ is sampled, followed by a duration $d$, and then $d$ observations are emitted using the state-specific emission parameters.

An HDP-HSMM models durations as state-specific but observation-independent. For a state $j$, the duration $d$ follows:

$$d - 1 \mid z = j \sim \mathrm{NB}(r_j, p_j) \tag{14}$$

where $p_j \in (0, 1)$ is a fixed scalar parameter.

The RED-HDP-HMM models durations dependent on the previous observation $\mathbf{y}_{t-1}$. The generative process is:

$$d - 1 \mid z = j, \mathbf{y}_{t-1} \sim \mathrm{NB}\left(r_j, \sigma\left(\mathbf{y}_{t-1}^\top \boldsymbol{\beta}_j^D + \beta_j^{D\mathrm{bias}}\right)\right) \tag{15}$$

We construct a RED-HDP-HMM that mimics the HDP-HSMM by choosing the following parameters for all states $j$:

$$\boldsymbol{\beta}_j^D = \mathbf{0} \quad \text{(the zero vector)} \tag{16}$$

$$\beta_j^{D\mathrm{bias}} = \mathrm{logit}(p_j) = \ln\left(\frac{p_j}{1 - p_j}\right) \tag{17}$$

Substituting these into the RED parameterization:

$$p(z = j, \mathbf{y}_{t-1}) = \sigma\left(\mathbf{y}_{t-1}^\top \mathbf{0} + \mathrm{logit}(p_j)\right) \tag{18}$$

$$= \sigma(\mathrm{logit}(p_j)) \tag{19}$$

$$= p_j \tag{20}$$

Thus, the duration distribution becomes $d - 1 \mid z = j \sim \mathrm{NB}(r_j, p_j)$. This distribution is independent of $\mathbf{y}_{t-1}$ and identical to the HDP-HSMM specification. Since the generative skeletons and all other components (transition kernels, emission densities) are structurally identical, the HDP-HSMM is a special case of the RED-HDP-HMM.

**Part 2: Strictness** We prove strictness by showing that RED-HDP-HMM allows for conditional dependence structures impossible in HDP-HSMM.

**2A: Structural Argument** In an HDP-HSMM, the duration $d_t$ is conditionally independent of the previous observation $\mathbf{y}_{t-1}$ given the current state $z_t$:

$$d_t \perp \mathbf{y}_{t-1} \mid z_t \quad \text{(HDP-HSMM)} \tag{21}$$

In a RED-HDP-HMM with $\boldsymbol{\beta}_j^D \neq \mathbf{0}$, the duration probability parameter $p(\mathbf{y}) = \sigma(\boldsymbol{\beta}_j^D \cdot \mathbf{y} + \beta_j^{D\mathrm{bias}})$ is a non-constant function of $\mathbf{y}$. Thus:

$$d_t \not\perp \mathbf{y}_{t-1} \mid z_t \quad \text{(RED-HDP-HMM)} \tag{22}$$

Since a non-constant function cannot be equal to a constant, no single scalar parameter $p_j$ can capture the distribution of a RED model where $\boldsymbol{\beta}_j^D \neq \mathbf{0}$. Furthermore, because the HDP-HSMM assumes a discrete set of states, it cannot approximate the continuous dependence of $d$ on $\mathbf{y}$ arbitrarily well without expanding the state space to infinity to partition the domain of $\mathbf{y}$.

**2B: Quantitative Separation** We construct a counterexample where the Total Variation (TV) distance between the RED model and *any* HDP-HSMM is strictly positive. *Setup:* Consider a 2-state model with 1D observations $y_t \in \mathbb{R}$.

- State 1 emissions: $y \mid z = 1 \sim \mathcal{N}(0, 1)$.

- State 2 RED parameters: $r_2 = 5$, $\beta_2^D = 1$, $\beta_2^{D\mathrm{bias}} = 0$.

The duration distribution for state 2 in the RED model is determined by $p(y) = \sigma(y)$.

- If $y_{t-1} = 2$, then $p \approx \sigma(2) \approx 0.88$. (Short durations expected).

- If $y_{t-1} = -2$, then $p \approx \sigma(-2) \approx 0.12$. (Long durations expected).

Let $P_{\text{RED}}$ denote the joint distribution of the RED model and $Q_{\text{HSMM}}$ denote the joint distribution of an arbitrary HDP-HSMM. The Total Variation distance between the joint distributions over a transition $(y_{t-1}, z_t = 2, d_t)$ is bounded by the integral of the conditional TV distances:

$$d_{\text{TV}}(P_{\text{RED}}, Q_{\text{HSMM}}) \geq \int_{\mathcal{Y}} P(y_{t-1}) \cdot d_{\text{TV}}\left(P(d \mid z = 2, y_{t-1}), Q(d \mid z = 2)\right) \, dy_{t-1} \tag{23}$$

In the HDP-HSMM, $Q(d \mid z = 2)$ is fixed to some $\text{NB}(r', p')$. In the RED model, the distribution $P(d \mid z = 2, y_{t-1})$ varies continuously with $y_{t-1}$. Because $P(d \mid z = 2, y_{t-1})$ is not constant with respect to $y$, there is no single distribution $Q(d)$ that satisfies $d_{\text{TV}}(P(\cdot|y), Q(\cdot)) = 0$ for all $y$. The integrand is strictly positive on a set of non-zero measure (since the Gaussian emission $P(y_{t-1})$ is positive everywhere). Therefore, $d_{\text{TV}}(P_{\text{RED}}, Q_{\text{HSMM}}) > 0$ for any choice of HDP-HSMM parameters. $\square$ $\square$

## C. Theorem 4: Forward-Backward Sampling Correctness

The forward-backward algorithm for sampling the joint state-duration sequence $(\mathbf{z}_{1:T}, \mathbf{d}_{1:T})$ is a key component of our Gibbs sampler. This section proves that the algorithm correctly samples from the conditional posterior.

**Theorem C.1** (Forward-Backward Sampling Correctness). *Let $D_{\max} \geq T$ be the maximum duration parameter, and let the initial state distribution be $P(z_1 = j) = \rho_j$ (the stationary distribution under ergodicity). For the **nonloopy** RED-HDP-HMM (where self-transitions are disallowed at segment boundaries, i.e., $\pi_{jj} = 0$) with **truncated duration support** $d \in \{1, \ldots, D_{\max}\}$, the forward-backward algorithm in Algorithm 1 (Step 1) produces exact samples from the conditional posterior:*

$$P(\mathbf{z}_{1:T}, \mathbf{d}_{1:T} \mid \mathbf{y}_{1:T}, \boldsymbol{\beta}, \bar{\pi}_j, \theta_j, \omega_j) \tag{24}$$

*under the following assumptions:*

1. ***Duration truncation**: The duration distribution $\text{NB}(d - 1 \mid r_j, p_j)$ is truncated to support $\{1, \ldots, D_{\max}\}$ and renormalized. This weak-limit approximation introduces negligible error when $D_{\max}$ exceeds the effective duration range (see Theorem 5.2).*

2. ***No left-censoring**: The sequence begins at the start of a fresh segment (i.e., $d_1$ is drawn from the full duration distribution, not a residual distribution).*

3. ***Right-censoring allowed**: The final segment may extend beyond time $T$ (i.e., $d_T \geq 1$ is not constrained to equal 1).*

*For autoregressive emissions where the likelihood depends on the previous observation, we define the shorthand $L_t(j) \triangleq P(\mathbf{y}_t \mid z_t = j, \mathbf{y}_{t-1})$. Since all observations are conditioned upon, these are computable scalar functions of the state $z_t$. The initial duration $d_1$ is sampled using only the bias term $\beta_{z_1}^{D\text{bias}}$ (since there is no preceding observation $\mathbf{y}_0$).*

*Proof.* We establish correctness by showing that the forward-backward decomposition correctly factors the joint posterior.
**Step 1: Joint Distribution Factorization** Unlike standard HMMs where the latent process is independent of observations, the RED-HDP-HMM includes autoregressive dependencies in both emissions and duration parameters. The joint distribution factors causally over time:

$$P(\mathbf{z}_{1:T}, \mathbf{d}1:T, \mathbf{y}1:T \mid \Theta) = \prod_{t=1}^{T} P(\mathbf{y}_t \mid z_t, \mathbf{y}_{t-1}) \cdot P(z_t, d_t \mid z_{t-1}, d_{t-1}, \mathbf{y}_{t-1}) \tag{25}$$

where $\Theta$ denotes all parameters. The emission term $L_t(z_t) = \mathcal{N}(\mathbf{y}_t \mid A_{z_t} \mathbf{y}_{t-1} + b_{z_t}, \Sigma_{z_t})$ acts as a potential function during inference since $\mathbf{y}_{1:T}$ is fully observed. **Step 2: Transition Structure** The transition kernel $P(z_t, d_t \mid z_{t-1}, d_{t-1}, \mathbf{y}_{t-1})$ is defined by the explicit duration mechanism. Given the nonloopy assumption ($\pi_{jj} = 0$), transitions are deterministic within a

segment and stochastic at boundaries:

$$P(z_t \mid z_{t-1}, d_{t-1}) = \begin{cases} \mathbf{1}[z_t = z_{t-1}] & \text{if } d_{t-1} > 1 \\ \pi_{z_{t-1}, z_t} & \text{if } d_{t-1} = 1 \quad (\text{where } \pi_{z_{t-1}, z_{t-1}} = 0) \end{cases} \tag{26}$$

$$P(d_t \mid z_t, d_{t-1}, \mathbf{y}_{t-1}) = \begin{cases} \mathbf{1}[d_t = d_{t-1} - 1] & \text{if } d_{t-1} > 1 \\ \text{NB}(d_t - 1 \mid r_{z_t}, p_{z_t}(\mathbf{y}_{t-1})) & \text{if } d_{t-1} = 1 \end{cases} \tag{27}$$

where $p_j(\mathbf{y}) = \sigma(\mathbf{y}^\top \boldsymbol{\beta}_j^D + \beta_j^{D\text{bias}})$ is the observation-dependent duration parameter. **Step 3: Forward Variables** Define the forward variable for state $j$ and remaining duration $d$ at time $t$ as the joint probability of the partial observation sequence and current latent state:

$$\alpha_t(j, d) = P(\mathbf{y}_{1:t}, z_t = j, d_t = d \mid \Theta) \tag{28}$$

**Base case** $(t = 1)$: We assume the observation sequence begins at the start of a fresh segment.

$$\alpha_1(j, d) = \rho_j \cdot \text{NB}(d - 1 \mid r_j, \sigma(\beta_j^{D\text{bias}})) \cdot L_1(j) \tag{29}$$

**Recursion** $(t > 1)$: We distinguish two cases based on the previous duration $d_{t-1}$. *Case 1: Duration continues* $(d_{t-1} > 1)$: Since $d_{t-1} > 1$ implies $z_t = z_{t-1}$ and $d_t = d_{t-1} - 1$, the only valid predecessor state for $(z_t = j, d_t = d)$ is $(j, d+1)$.

$$\alpha_t^{\text{cont}}(j, d) = \alpha_{t-1}(j, d+1) \cdot L_t(j) \tag{30}$$

*Case 2: New segment starts* $(d_{t-1} = 1)$: If $d_{t-1} = 1$, a state transition occurs. The predecessor must be some state $k$ with remaining duration 1. Due to the nonloopy constraint, $k \neq j$.

$$\alpha_t^{\text{switch}}(j, d) = \sum_{k \neq j} \alpha_{t-1}(k, 1) \cdot \pi_{k,j} \cdot \text{NB}(d - 1 \mid r_j, p_j(\mathbf{y}_{t-1})) \cdot L_t(j) \tag{31}$$

**Combined recursion**:

$$\alpha_t(j, d) = L_t(j) \cdot \left[ \alpha_{t-1}(j, d+1) \cdot \mathbf{1}[d + 1 \leq D_{\max}] \qquad + \sum_{k \neq j} \alpha_{t-1}(k, 1) \cdot \pi_{k,j} \cdot \text{NB}(d - 1 \mid r_j, p_j(\mathbf{y}_{t-1})) \right]$$

**Step 4: Backward Sampling** The backward pass samples $(z_t, d_t)$ for $t = T, T-1, \ldots, 1$. At $t = T$, we sample $(z_T, d_T)$ from the normalized distribution:

$$P(z_T, d_T \mid \mathbf{y}_{1:T}) = \frac{\alpha_T(z_T, d_T)}{\sum_{j,d} \alpha_T(j, d)} \tag{32}$$

For $t < T$, we sample the predecessor $(z_t, d_t)$ given the successor $(z_{t+1}, d_{t+1})$. By Bayes' rule and the Markov property:

$$P(z_t, d_t \mid z_{t+1}, d_{t+1}, \mathbf{y}_{1:T}, \Theta) \propto \alpha_t(z_t, d_t) \cdot P(z_{t+1}, d_{t+1} \mid z_t, d_t, \mathbf{y}_t) \tag{33}$$

The term $L_{t+1}(z_{t+1})$ is omitted as it is constant for a fixed successor. The set of valid predecessors is sparse: **Case 1: Continuation Predecessor** (Valid if $d_{t+1} < D_{\max}$)

- **Candidate**: $(z_t, d_t) = (z_{t+1}, d_{t+1} + 1)$

- **Weight**: $w_{\text{cont}} \propto \alpha_t(z_{t+1}, d_{t+1} + 1)$

**Case 2: Switch Predecessors** (Valid for all $k \neq z_{t+1}$) The nonloopy assumption ensures we only consider $k \neq z_{t+1}$, matching the sparse structure of the forward pass.

- **Candidates**: $(z_t, d_t) = (k, 1)$ for each $k \in \{1, \ldots, L\} \setminus \{z_{t+1}\}$

- **Weight**: $w_{\text{switch}}(k) \propto \alpha_t(k, 1) \cdot \pi_{k, z_{t+1}} \cdot \text{NB}(d_{t+1} - 1 \mid r_{z_{t+1}}, p_{z_{t+1}}(\mathbf{y}_t))$

**Step 5: Correctness via Markov Property** The validity relies on the conditional independence:

$$P(z_{t+2:T}, \mathbf{d}t + 2 : T \mid zt + 1, d_{t+1}, \mathbf{y}_{1:T}) = P(z_{t+2:T}, \mathbf{d}t + 2 : T \mid zt + 1, d_{t+1}, \mathbf{y}_{t+1:T}) \tag{34}$$

Conditioning on $(z_{t+1}, d_{t+1})$ d-separates future latent variables from past latent variables. Although parameters depend on observations $\mathbf{y}$, the full observation sequence is fixed, preserving the validity of the d-separation. **Step 6: Computational Complexity** The forward pass computes $\alpha_t(j, d)$ for all states and durations:

- Complexity: $O(T \cdot L \cdot (L + D_{\max}))$

The backward sampling pass is highly efficient due to sparsity. For a fixed successor, we only evaluate 1 continuation candidate and $L - 1$ switch candidates.

- Complexity: $O(T \cdot L)$

**Conclusion** The algorithm correctly samples from $P(\mathbf{z}_{1:T}, \mathbf{d}_{1:T} \mid \mathbf{y}_{1:T}, \Theta)$ by combining the forward marginal probabilities with the exact conditional inverse transitions. $\square$

# D. Prior Distributions

This section documents the prior distributions used in our implementation, which follows a fully Bayesian approach with conjugate or semi-conjugate priors for efficient Gibbs sampling.

## D.1. AR Emission Priors

The AR coefficients and innovation covariances follow a Matrix Normal-Inverse Wishart (MNIW) prior, the conjugate prior for multivariate linear regression:

$$(A_j, \Sigma_j) \sim \text{MNIW}(M_0, V_0, S_0, n_0) \tag{35}$$

where:

- $M_0 \in \mathbb{R}^{M \times (M+1)}$ is the prior mean for $[A_j \mid b_j]$ (default: $M_0 = 0$)

- $V_0 \in \mathbb{R}^{(M+1) \times (M+1)}$ is the prior covariance for the regression coefficients (default: $V_0 = I$)

- $S_0 \in \mathbb{R}^{M \times M}$ is the prior scale matrix for $\Sigma_j$ (default: $S_0 = 0.75 \times \hat{\Sigma}_{\Delta \mathbf{y}}$, where $\hat{\Sigma}_{\Delta \mathbf{y}}$ is the empirical covariance of observation differences)

- $n_0 > M - 1$ is the prior degrees of freedom (default: $n_0 = M + 2$)

The posterior for each state $j$ is also MNIW, updated using sufficient statistics:

$$\bar{Y}_{t-1}^\top \bar{Y}_{t-1} = \sum_{t:z_t=j} \tilde{\mathbf{y}}_{t-1} \tilde{\mathbf{y}}_{t-1}^\top \quad (\text{with } \tilde{\mathbf{y}}_{t-1} = [\mathbf{y}_{t-1}^\top, 1]^\top) \tag{36}$$

$$Y_t^\top \bar{Y}_{t-1} = \sum_{t:z_t=j} \mathbf{y}_t \tilde{\mathbf{y}}_{t-1}^\top \tag{37}$$

## D.2. Duration Priors

For the Negative Binomial duration model with regression:

**Dispersion parameter**: $r_j \sim \text{Gamma}(a_0, g_0)$ with $a_0 = g_0 = 0.01$ (weakly informative).

**Regression coefficients**: $\boldsymbol{\beta}_j^D \sim \mathcal{N}(0, \alpha_j I)$ where $\alpha_j$ controls sparsity.

**Sparsity parameters**: $\alpha_j \sim \text{Gamma}(c_0, d_0)$ with $c_0 = d_0 = 0.01$.

**Precision**: $\phi \sim \text{Gamma}(e_0, f_0)$ with $e_0 = f_0 = 0.01$.

## D.3. HDP Concentration Parameters

The Dirichlet process concentration parameters have Gamma hyperpriors:

$$\alpha_0 \sim \text{Gamma}(a_\alpha, b_\alpha) \quad \text{(transition concentration, default: } a_\alpha = 1, b_\alpha = 0.01) \tag{38}$$

$$\gamma_0 \sim \text{Gamma}(a_\gamma, b_\gamma) \quad \text{(base measure concentration, default: } a_\gamma = 2, b_\gamma = 1) \tag{39}$$

These are sampled using auxiliary variable methods following Teh et al. (2006).

## D.4. Stickiness Parameters

For the sticky HDP-HMM variants, the stickiness parameter $\kappa$ (or its functional equivalent) is sampled using a grid-based posterior approximation over the range $[0.01, 0.99]$ with 30 grid points.

## D.5. Pólya-Gamma Augmentation for Duration Inference

The Negative Binomial regression model for durations uses a logistic link function:

$$d_t - 1 \sim \text{NB}(r_{z_t}, p_t), \quad \text{logit}(p_t) = \psi_t = \mathbf{y}_{t-1}^\top \boldsymbol{\beta}_{z_t}^D + \beta_{z_t}^{D\text{bias}} \tag{40}$$

Direct sampling of $(\boldsymbol{\beta}_j^D, \beta_j^{D\text{bias}})$ is intractable due to the nonlinear sigmoid link. Following Polson et al. (2013), we introduce Pólya-Gamma auxiliary variables to obtain conditionally Gaussian posteriors.

**Key identity** (Polson et al., 2013): For any $a > 0$ and $b \in \mathbb{R}$:

$$\frac{e^{a\psi}}{(1 + e^\psi)^b} = 2^{-b} e^{\kappa\psi} \int_0^\infty e^{-\omega\psi^2/2} \, p(\omega) \, d\omega \tag{41}$$

where $\kappa = a - b/2$ and $\omega \sim \text{PG}(b, 0)$ is a Pólya-Gamma random variable.

**Application to NB likelihood**: The Negative Binomial PMF can be written as:

$$P(d \mid r, p) \propto p^r (1 - p)^d = \frac{e^{r\psi}}{(1 + e^\psi)^{r+d}} \tag{42}$$

where $\psi = \text{logit}(p)$. Setting $a = r$ and $b = r + d$ in the key identity yields an augmented likelihood that is conditionally Gaussian in $\psi$.

**Conditional distributions**:

*(a) Sampling $\omega_{j,t}$*: For each segment start time $t$ with $z_t = j$, the auxiliary variable follows:

$$\omega_{j,t} \mid d_t, r_j, \psi_t \sim \text{PG}(r_j + d_t, \psi_t) \tag{43}$$

where $\psi_t = \mathbf{y}_{t-1}^\top \boldsymbol{\beta}_j^D + \beta_j^{D\text{bias}}$. This is Step 2 of Algorithm 1.

*(b) Sampling $(\boldsymbol{\beta}_j^D, \beta_j^{D\text{bias}})$*: Given $\{\omega_{j,t}\}$, the complete-data log-likelihood becomes quadratic in the regression coefficients. Let $\tilde{\mathbf{y}}_{t-1} = (\mathbf{y}_{t-1}^\top, 1)^\top$ and $\tilde{\boldsymbol{\beta}}_j = ((\boldsymbol{\beta}_j^D)^\top, \beta_j^{D\text{bias}})^\top$. With Gaussian prior $\tilde{\boldsymbol{\beta}}_j \sim \mathcal{N}(0, (\alpha^D)^{-1}I)$, the posterior is:

$$\tilde{\boldsymbol{\beta}}_j \mid \{\omega_{j,t}\}, \{d_t\}, \mathbf{y} \sim \mathcal{N}(m_n, \Lambda_n^{-1}) \tag{44}$$

$$\Lambda_n = \alpha^D I + \sum_{t \in S_j} \omega_{j,t} \tilde{\mathbf{y}}_{t-1} \tilde{\mathbf{y}}_{t-1}^\top \tag{45}$$

$$m_n = \Lambda_n^{-1} \sum_{t \in S_j} \kappa_{d_t} \tilde{\mathbf{y}}_{t-1} \tag{46}$$

where $S_j$ is the set of segment start times with state $j$ and $\kappa_d = (r - d)/2$.

*(c) Sampling $r_j$:* Following the compound Poisson representation of the Negative Binomial (Zhou & Carin, 2012), we introduce auxiliary count variables $\ell_t \sim \mathrm{CRT}(d_t, r_j)$ (Chinese Restaurant Table distribution). The posterior for $r_j$ is then:

$$r_j \mid \{\ell_t\}, \{p_t\} \sim \Gamma\left(a_0 + \sum_{t \in S_j} \ell_t, \ g_0 - \sum_{t \in S_j} \log(1 - p_t)\right) \tag{47}$$

This derivation justifies the auxiliary variable sampling in Algorithm 1 and ensures that all Gibbs updates have closed-form conditional distributions.

### D.6. Wallclock Timings and MCMC Diagnostics

This section reports the additional Bee Dance diagnostics requested by the reviewers. The wallclock measurements in Table 3 should be interpreted as approximate order-of-magnitude comparisons: the runs may differ in hardware, core count, checkpointing cadence, and filesystem load, and scheduler queue time is excluded.

*Table 3.* Approximate wallclock timing comparison. Times are measured inside the Python runners, exclude scheduler queue time, and are only directly comparable for runs made on similar hardware with the same number of cores and checkpointing cadence.

| Dataset | Model | Runs | Cores | Wallclock (min) | Sec./iter. |
|---|---|---|---|---|---|
| Bee Dance | HDP-HSMM (NB) | 3 | 2 | $833.5 \pm 1.4$ | $10.002 \pm 0.017$ |
| Bee Dance | HDP-HSMM (Poisson) | 3 | 2 | $806.7 \pm 0.8$ | $9.680 \pm 0.009$ |
| Bee Dance | R-DS-HDP-HMM | 3 | 2 | $414.3 \pm 1.4$ | $4.972 \pm 0.017$ |
| Bee Dance | RED-HDP-HMM (Loopy) | 3 | 2 | $1391.8 \pm 23.9$ | $16.702 \pm 0.287$ |
| Bee Dance | RED-HDP-HMM (Nonloopy) | 3 | 2 | $1260.6 \pm 21.6$ | $15.127 \pm 0.260$ |

RED-HDP-HMM is more expensive per iteration on Bee Dance than the recurrent sticky baseline, as expected from the explicit duration dimension in the forward-backward pass. The nonloopy and loopy RED-HDP-HMM variants require approximately 15.1 and 16.7 seconds per iteration, respectively, compared with 5.0 seconds for RS-HDP-HMM and 9.7–10.0 seconds for the two HDP-HSMM baselines. This overhead is the computational cost of modeling non-geometric, observation-dependent durations explicitly.

Table 4 reports effective sample sizes (ESS) for scalar chain summaries. ESS is strong for the concentration parameters $\alpha_0$ and $\gamma_0$, indicating stable sampling of the main HDP regularization parameters. Some chain-specific summaries mix more slowly, especially the duration-coefficient norm $\|\xi\|_2$ and a few nonloopy summaries. We therefore treat the scalar diagnostics as evidence of usable but nontrivial posterior geometry rather than as evidence that every parameter mixes equally quickly.

*Table 4.* Effective sample size (ESS) for scalar MCMC diagnostics. ESS is computed per chain from the positive initial sequence of autocorrelations and summarized across chains.

| Dataset | Model | Quantity | Chains | Mean ESS | Min ESS |
|---|---|---|---|---|---|
| Bee Dance | R-DS-HDP-HMM | Test log likelihood | 3 | 231.8 | 15.4 |
| Bee Dance | RED-HDP-HMM (Loopy) | Test log likelihood | 3 | 415.4 | 263.0 |
| Bee Dance | RED-HDP-HMM (Loopy) | Occupied states | 3 | 66.6 | 52.5 |
| Bee Dance | RED-HDP-HMM (Loopy) | $\alpha_0$ | 3 | 921.3 | 900.2 |
| Bee Dance | RED-HDP-HMM (Loopy) | $\gamma_0$ | 3 | 2987.2 | 2250.6 |
| Bee Dance | RED-HDP-HMM (Loopy) | $\|\xi\|_2$ | 3 | 128.2 | 3.6 |
| Bee Dance | RED-HDP-HMM (Nonloopy) | Test log likelihood | 3 | 334.7 | 4.0 |
| Bee Dance | RED-HDP-HMM (Nonloopy) | Occupied states | 3 | 39.4 | 5.3 |
| Bee Dance | RED-HDP-HMM (Nonloopy) | $\alpha_0$ | 3 | 1349.9 | 1301.9 |
| Bee Dance | RED-HDP-HMM (Nonloopy) | $\gamma_0$ | 3 | 1604.6 | 897.0 |
| Bee Dance | RED-HDP-HMM (Nonloopy) | $\|\xi\|_2$ | 3 | 3.6 | 3.3 |

To assess whether independent chains recover comparable segmentations, Table 5 compares pairwise MAP state sequences after optimal label alignment with the Hungarian algorithm. RED-HDP-HMM (Nonloopy) gives the most stable segmentation among the evaluated methods, with mean pairwise Hamming distance 0.054 and maximum 0.070. This is lower than loopy RED-HDP-HMM (mean 0.138), RS-HDP-HMM (mean 0.109), HDP-HSMM with Negative Binomial durations (mean 0.339), and HDP-HSMM with Poisson durations (mean 0.418). Thus, the nonloopy explicit-duration construction substantially reduces segmentation-level chain variability on Bee Dance.

Table 6 examines a stricter question: whether the duration regression coefficients $\xi$ themselves are stable after state-label alignment. The answer is mixed. Loopy RED-HDP-HMM has moderate across-chain coefficient variability, while nonloopy

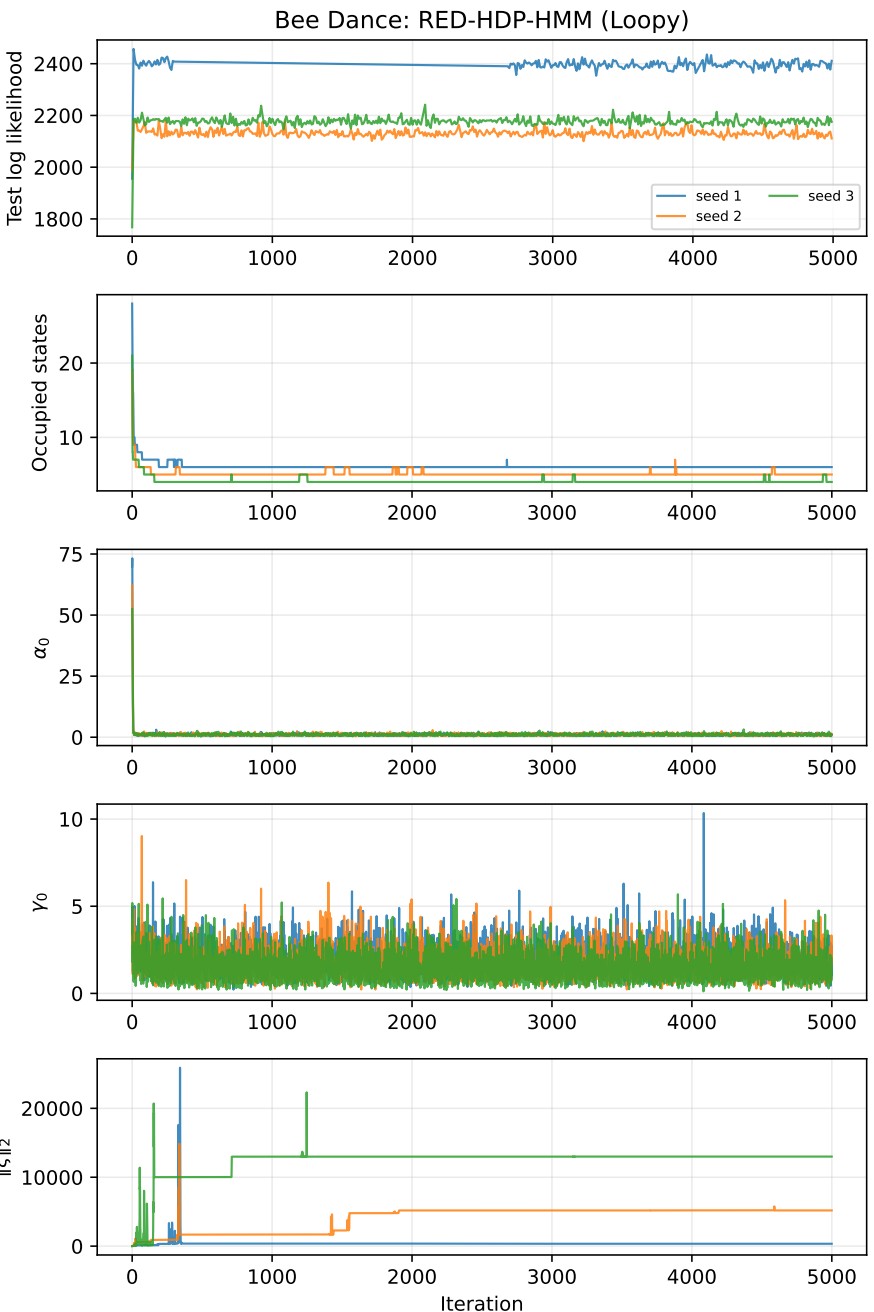

*Figure 5.* Trace diagnostics for loopy RED-HDP-HMM on Bee Dance. The panels show test log likelihood, occupied states, HDP concentration parameters, and the duration-coefficient norm $\|\xi\|_2$.

*Table 5.* Pairwise MAP state-sequence distances across chains after optimal label alignment with the Hungarian algorithm. Low distances indicate convergence to equivalent segmentations up to label permutation.

| Dataset | Model | Chains | Pairs | Mean | Max |
|---|---|---|---|---|---|
| Bee Dance | HDP-HSMM (NB) | 3 | 3 | 0.339 | 0.366 |
| Bee Dance | HDP-HSMM (Poisson) | 3 | 3 | 0.418 | 0.462 |
| Bee Dance | R-DS-HDP-HMM | 3 | 3 | 0.109 | 0.148 |
| Bee Dance | RED-HDP-HMM (Loopy) | 3 | 3 | 0.138 | 0.170 |
| Bee Dance | RED-HDP-HMM (Nonloopy) | 3 | 3 | 0.054 | 0.070 |

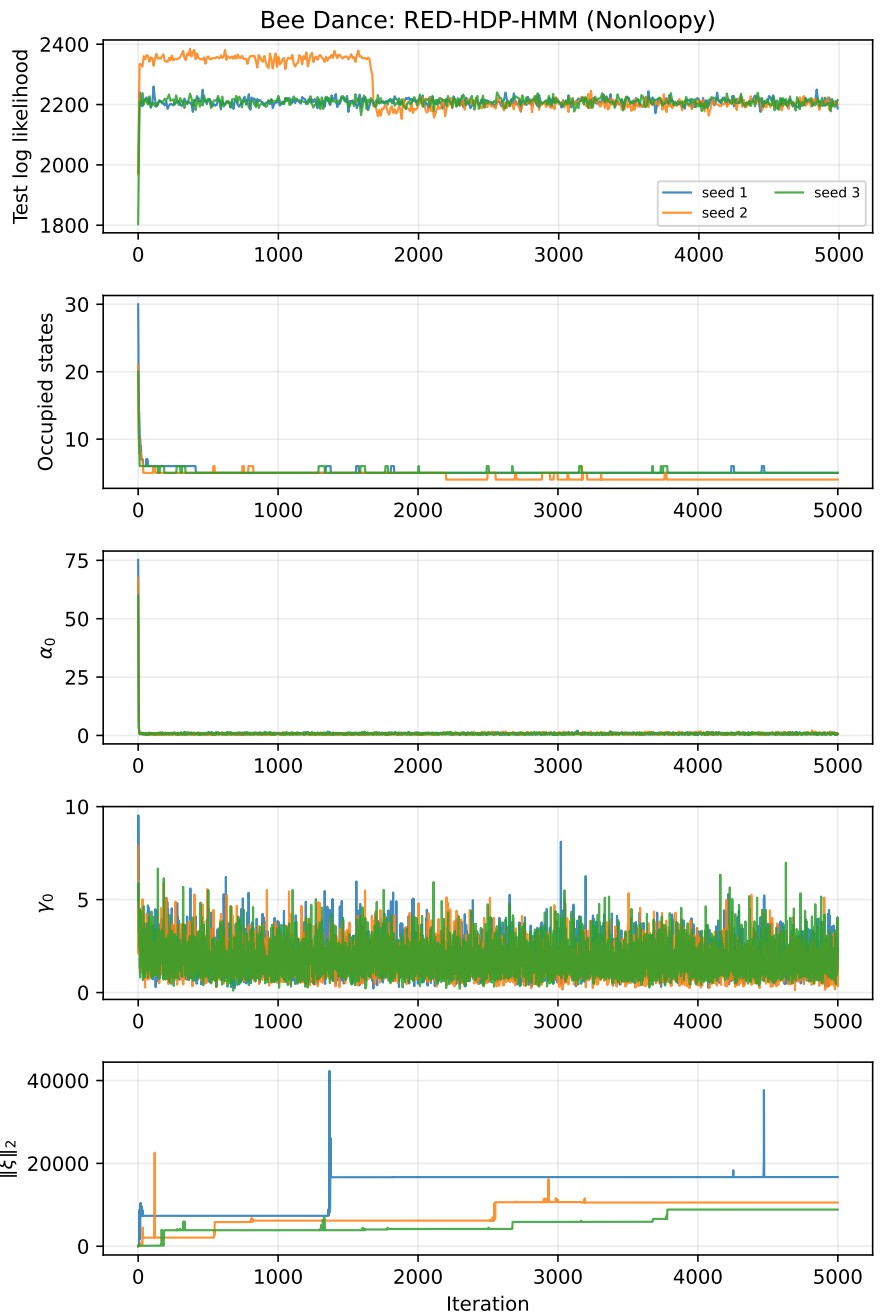

*Figure 6.* Trace diagnostics for nonloopy RED-HDP-HMM on Bee Dance. The panels show test log likelihood, occupied states, HDP concentration parameters, and the duration-coefficient norm $\|\xi\|_2$.

RED-HDP-HMM shows large variability in some aligned coefficients despite its highly stable MAP segmentations. This pattern is consistent with residual weak identification in the emission-duration parameterization: different coefficient values can support similar state sequences. Consequently, we interpret the main empirical result as strong evidence for stable segmentation and useful observation-dependent duration modeling, but we recommend caution when assigning mechanistic meaning to individual state-specific duration coefficients.

*Table 6.* Across-chain consistency of duration regression coefficients $\xi$ after state-label alignment.

| Dataset | Model | Chains | States | Mean coord. SD | Max coord. SD |
|---------|-------|--------|--------|----------------|---------------|
| Bee Dance | RED-HDP-HMM (Loopy) | 3 | 6 | 0.922 | 9.55 |
| Bee Dance | RED-HDP-HMM (Nonloopy) | 3 | 6 | 15.6 | 453 |

## E. Discussion

The theorems and corollary established above provide the theoretical foundation for the RED-HDP-HMM:

1. **Theorem 5.1 (State Truncation Error)** quantifies the approximation quality when restricting to $L$ latent states. The bound $d_{\mathrm{TV}}(\Pi_L^{(T)}, \Pi_\infty^{(T)}) \leq T \cdot e^{-c(L-1)}$ provides guidance for choosing $L$ in practice: truncation levels $L = 30$–$50$ yield negligible error for sequences up to $T \sim 10^4$.

2. **Theorem 5.2 (Duration Truncation Error)** bounds the approximation quality when restricting durations to $D_{\max}$ via the clamped model. The bound $d_{\mathrm{TV}}(\Pi_{D_{\max}}^{(T)}(\cdot \mid \Theta), \Pi^{(T)}(\cdot \mid \Theta)) \leq C(R, \epsilon) \cdot T \cdot e^{-\lambda(\epsilon)D_{\max}}$ shows that duration truncation introduces exponentially small error, with $D_{\max} = 100$–$200$ sufficient for typical applications.

3. **Corollary 5.3 (Joint Truncation)** combines the state and duration truncation bounds via the triangle inequality, giving a complete characterization of the weak-limit approximation error for the $(L, D_{\max})$-truncated model.

4. **Theorem 5.4 (Expressiveness)** demonstrates that RED-HDP-HMM is a genuine extension of HDP-HSMM, capable of capturing observation-dependent duration patterns that are fundamentally inaccessible to models with state-specific but observation-independent durations.

5. **Theorem C.1 (Forward-Backward Sampling Correctness)** proves that the dynamic programming algorithm for sampling latent state-duration sequences produces exact samples from the conditional posterior. This extends beam sampling ideas to explicit duration models with observation-dependent durations.

Together, these results establish RED-HDP-HMM as a well-founded model with rigorous theoretical guarantees, complementing the empirical results presented in the main paper.

