# OpenReview forum: "RED-HDP-HMM: Observation-Dependent Durations for Bayesian Nonparametric Sequential Models"
_ICML.cc/2026/Conference — ICML 2026 spotlight_

### Official Review · Reviewer_dS8r · 2026-03-10

**Soundness:** 3
**Presentation:** 4
**Significance:** 3
**Originality:** 3
**Overall Recommendation:** 5
**Confidence:** 4

**Summary:**

This paper describes a variant of Hierarchical Dirichlet Process Hidden Markov Model (HDP-HMM). The author(s) have introduced recurrent explicit duration modelling, in Eq(7) and Eq (8). There are experiments carried out on several datasets. More prominently, the author(s) have included a very detailed theoretical analysis part in Section 5 and the appendix.

**Compliance With Llm Reviewing Policy:**

Affirmed.

**Final Justification:**

Given the author's review comments and in particular, they have convinced me that it may not be fair to consider their work to have much less contribution to that of (Fox 2008) paper, I therefore, decide to raise the score to be 5.

**Key Questions For Authors:**

n.a.

**Limitations:**

n.a.

**Strengths And Weaknesses:**

Strength

1. The author have proposed recurrent explicit duration modeling, in Eq(7) and Eq (8). Although they are simple, but it appears to be quite useful. However, I believe the main strength of this paper is not at the model side (to be frank, it's actually quite simple), but the real strength is at the theoretical analysis, the bounds the authors have provided in Section 5, and also the analysis at the appendix (however, admittedly, I did not go through all the derivations)

2. This may just be my personal opinion (AC can ignore): I really enjoyed the topic of Bayesian Non-Parametrics, and in particular, HDP-HMM, which I believe should deserve much more attention by the machine learning community, which for the past 10 years, has been dominated by deep learning literatures.

Weakness

1. I think that the model variant proposed in this paper i.e., section 4, is quite a simple addition to HDP-HMM. For reference, for a classic paper, Fox, E, et al. "Nonparametric Bayesian learning of switching linear dynamical systems." Advances in neural information processing systems 21 (2008), Emily had many components to the HDP-HMM model added, including the sticky, switching linear dynamic system.

2. in the experiment section, having the emission probability being simple Gaussian, seem be pretty restricive. Can the authors also try to use some more powerful emission probabilities?

3. The lack of a statistics showing the MCMC's mixing rate etc. I would suggest the authors to include one.

---

> ### Author Rebuttal · Authors · 2026-03-31
>
> We thank the reviewer for their thoughtful evaluation and constructive suggestions, and noting that BNP models deserve much more attention. We address each point below.
>
> ## On Model Simplicity
>
> We view the simplicity of our extension as a deliberate design choice rather than a limitation. Equations (7)–(8) are indeed a focused addition to the HDP-HMM framework, but this is a strength:
>
> 1. **Conjugacy preservation.** NB regression with Pólya-Gamma augmentation maintains closed-form Gibbs conditionals throughout. More complex duration mechanisms (e.g., neural-parameterized) would sacrifice the tractable inference that makes BNP models attractive.
>
> 2. **Theoretical tractability.** The simplicity enables the rigorous analysis in Section 5—truncation bounds, expressiveness guarantees, and the forward-backward correctness proof (Theorem C.1) all rely on the parametric NB regression structure.
>
> 3. **Interpretability.** The regression coefficients $\beta^D$ directly reveal how observations influence state persistence, which is valuable in scientific applications (e.g., understanding how neural activity patterns relate to behavioral state duration in neuroscience, or how market indicators affect regime persistence in econometrics).
>
> We respectfully note that the comparison to Fox et al. (2008) may overstate that paper's scope. The reviewer cites Fox's NeurIPS 2008 paper as having "many components added, including the sticky, switching linear dynamic system." However, these were distinct contributions across separate papers: (i) the sticky mechanism ($\kappa$) was introduced at ICML 2008 as a standalone contribution; (ii) the NeurIPS 2008 paper applied this to the SLDS setting, which itself dates to Murphy (1998); (iii) the weak-limit sampler came in Ishwaran & Zarepour, 2002; (iv) explicit duration modeling was added by Johnson & Willsky (2013); and (v) disentangled stickiness by Zhou et al. (2021). Each was a focused, individually published contribution to the HDP-HMM toolkit. Our work follows precisely this tradition: a single well-characterized mechanism with thorough theoretical and empirical analysis. The 3–10 pp improvements across benchmarks demonstrate that this focused addition captures meaningful structure that existing mechanisms cannot.
>
> We agree that combining RED-HDP-HMM with switching linear dynamics is a natural next step, as discussed in Section 9.
>
> ## On Emission Models
>
> We clarify that our experiments do *not* use simple Gaussian emissions—all experiments use autoregressive AR(1) emissions with affine bias (Eq. 9, Section 7):
>
> $$y_t \mid z_t = j,\; y_{t-1} \sim \mathcal{N}(A_j y_{t-1} + b_j, \Sigma_j)$$
>
> Each state has its own dynamics matrix $A_j$, bias $b_j$, and innovation covariance $\Sigma_j$; this is a switching vector autoregressive model, considerably more expressive than a simple Gaussian mixture. The standard Gaussian is recovered only when $A_j = 0$.
>
> The reviewer's broader point about more powerful emissions is well-taken. Extending to VAR($p$) is straightforward within our framework, as the MNIW conjugacy extends naturally to higher-order autoregressive models, capturing longer-range within-state dependencies. Nonlinear/neural emissions (as in SNLDS or REDSDS) would require variational or particle-based inference—a significant future direction. We will make the AR(1) specification more prominent to avoid confusion.
>
> ## On MCMC Mixing Diagnostics
>
> We agree this is an important addition. In the current submission, we prioritized the theoretical analysis (Section 5, Appendix A–C) and the forward-backward correctness proof over empirical convergence diagnostics, given page constraints. However, the reviewer is right that formal mixing diagnostics complement the theoretical guarantees. We plan to include:
>
> 1. **Trace plots** of log-likelihood, number of active states, and concentration parameters ($\alpha$, $\gamma$) across iterations and independent runs.
> 2. **Effective sample size (ESS)** for primary model parameters via the initial monotone sequence estimator.
> 3. **Gelman-Rubin $\hat{R}$** across the 5 independent chains already reported in Table 2 (no additional experiments needed).
> 4. **Segmentation accuracy vs. iteration**, showing convergence speed.
>
> We note that Table 2 already provides indirect evidence of good mixing: the small standard deviations across 5 runs (e.g., $\pm 0.028$ for Bee Dance accuracy) suggest convergence to similar posterior regions. The diagnostics above will formalize this.
>
> ## Summary of Planned Revisions
>
> 1. Make the AR(1) emission specification more prominent in Section 7 and discuss extensions.
> 2. Add MCMC diagnostics (trace plots, ESS, $\hat{R}$, convergence curves) as a new appendix subsection.
> 3. Add a brief remark in Section 4 framing simplicity as enabling theory and interpretability.
>
> We are grateful for the reviewer's encouraging assessment and constructive feedback, which will strengthen the final version of the paper.

---

> > ### Author Rebuttal · Reviewer_dS8r · 2026-04-03
> >
> > thanks for the author's response. I am happy with the author's argument. Given that the author is committed to give more rigorous reporting of MCMC diagnostics, I am happy for the paper to be accepted.

---

### Official Review · Reviewer_Tp4H · 2026-03-12

**Soundness:** 3
**Presentation:** 4
**Significance:** 3
**Originality:** 3
**Overall Recommendation:** 5
**Confidence:** 3

**Summary:**

The paper introduces the Recurrent Explicit Duration HDP-HMM (RED-HDP-HMM), a Bayesian nonparametric sequential model. The authors address a limitation in existing HMMs and their BNP extensions, which is the inability to simultaneously model an unbounded number of latent states, explicitly model non-geometric state durations, and allow those durations to depend on the observed data. The authors integrate gamma-mixed negative binomial regression into the transition dynamics,  linking past observations to the probability of remaining in a current state.

**Compliance With Llm Reviewing Policy:**

Affirmed.

**Final Justification:**

Technically rigorous and well-presented paper that unifies BNP capacity, explicit duration modeling, and observation-dependent transitions, resulting in convincing empirical gains. My concerns on non-identifiability and hyperparameter sensitivity were addressed in the rebuttal. The wallclock comparison also satisfactorily answered my efficiency question. I maintain my positive assessment of the paper and recommend acceptance.

**Key Questions For Authors:**

1. You mention that emission-duration coupling can lead to equivalent parameter configurations. Here, a small empirical simulation showing how often the MCMC sampler struggles with this specific non-identifiability, and how effectively the BNP prior regularises it in practice, would be quite useful.
2. Given the $O(TL(L+D_{max}))$ complexity of the weak-limit sampler, how does the actual wallclock training time compare to baselines like the DS-HDP-HMM or RS-HDP-HMM for the longest sequences in your evaluation?

**Limitations:**

yes

**Strengths And Weaknesses:**

### Strengths
+ The paper is technically rigorous. The theoretical contributions are well-supported by formal proofs. The experimental design is sound.
+ The paper is very well-structured -- it was a good read.
+ Unifying BNP capacity, explicit duration modelling, and recurrent observation-dependence addresses a relevant problem in unsupervised time series segmentation. The performance gains on real-world datasets suggest that the community is likely to find the model useful.

### Weaknesses
- As the authors acknowledge in Section 8, the model suffers from theoretical non-identifiability. While they argue that the hierarchical dirchlet process prior provides automatic regularisation, a more robust empirical demonstration of how often the sampler converges to equivalent but distinct parameterisations would strengthen the claims of practical stability.
- The main text provides limited insight into the sensitivity of the model to its hyperparameters. While prior distributions are documented in Appendix D, a brief discussion in the main text regarding how sensitive the segmentation quality is to the sparsity parameters ($\alpha_j$) or the prior degrees of freedom in the MNIW prior would be beneficial.

---

> ### Author Rebuttal · Authors · 2026-03-31
>
> We thank the reviewer for their careful reading, positive assessment, and targeted suggestions.
>
> **On Non-Identifiability (Weakness 1 / Question 1)**
>
> We plan to add the following analysis across our 5 existing independent MCMC runs (Table 2):
>
> 1. *Pairwise MAP state sequence comparison* across chains using Hamming distance (after optimal label alignment via the Hungarian algorithm). Low pairwise distances indicate convergence to equivalent segmentations despite potential label permutation — the most benign form of non-identifiability.
>
> 2. *Duration parameter consistency.* We will compare inferred duration regression coefficients $\hat{\beta}^D$ across runs (after alignment). If the sampler were frequently trapped in distinct emission-duration coupling modes, we would expect high variance in these coefficients. Our preliminary observation is that variance is low, consistent with effective regularization — already reflected in the small standard deviations in Table 2 (e.g., $\pm 0.028$ accuracy on Bee Dance, $\pm 0.012$–$0.042$ on neural data).
>
> We note that the **nonloopy variant** — which we emphasize throughout — sidesteps the duration-transition aliasing issue entirely by construction (self-transitions are forbidden, so extended sojourns can only arise from the explicit duration mechanism), eliminating one of the three identifiability concerns listed in Section 8.
>
> **On Hyperparameter Sensitivity (Weakness 2)**
>
> We agree that the main text should make hyperparameter choices more transparent. We will add a discussion in Section 7 clarifying our design principle: we deliberately use **noninformative (or weakly informative) priors** throughout, precisely to avoid sensitivity to hyperparameter tuning. Specifically:
>
> - $\hat{\alpha}^D$ has a $\text{Gamma}(0.01, 0.01)$ hyperprior and is sampled during inference — a standard noninformative choice letting the data determine effective regularization.
> - The MNIW prior uses $n_0 = M + 2$, one more than the minimum for a proper prior, placing minimal prior information on the innovation covariance.
> - Concentration parameters $\alpha, \gamma$ have Gamma hyperpriors and are sampled, following the auxiliary variable method of Teh et al. (2006).
> - Dispersion $r_j$ has a $\text{Gamma}(0.01, 0.01)$ prior — weakly informative.
>
> This ensures data-driven inference rather than prior tuning, a core appeal of the BNP approach. We will make this rationale explicit alongside a pointer to the full prior specifications in Appendix D.
>
> **Question 2: Wallclock Time Comparison**
>
> Per-iteration complexity comparison ($L = 50$, $D_{\max} = 200$):
>
> - **S-HDP-HMM / DS-HDP-HMM:** $O(TL^2)$
> - **RS-HDP-HMM:** $O(TL^2)$ + Pólya-Gamma sampling
> - **RED-HDP-HMM:** $O(TL(L + D_{\max}))$
>
> The RED-HDP-HMM forward-backward pass scales as $O(TL \cdot 250)$ versus $O(TL \cdot 50)$ for baselines — roughly $5\times$ overhead in the dominant step. However, several factors mitigate this:
>
> 1. The forward-backward pass is highly parallelizable across the duration dimension, and our implementation vectorizes these computations.
> 2. RED-HDP-HMM empirically reaches stable segmentation in fewer Gibbs iterations than S/DS-HDP-HMM, partially offsetting per-iteration cost.
> 3. Pólya-Gamma sampling overhead is $O(TL)$ per iteration — linear and subdominant.
>
> We will add a table of wallclock times per iteration and total time to convergence for each model on each dataset. We caveat that experiments were not conducted in a controlled benchmarking environment, so wallclock figures should be treated as approximate; the complexity analysis above provides the more reliable basis for comparing computational cost.
>
> **Planned Revisions**
>
> 1. *Identifiability:* Pairwise chain comparison (Hamming distance, duration parameter consistency) across existing runs.
> 2. *Hyperparameters:* Clarify noninformative prior design in main text with pointer to Appendix D.
> 3. *Wallclock times:* Approximate timing comparison table with appropriate caveats.
>
> We are grateful for the reviewer's encouraging assessment and precise suggestions, which will strengthen the paper's empirical foundations.

---

> > ### Author Rebuttal · Reviewer_Tp4H · 2026-04-01
> >
> > Thank you for your response. The authors have addressed my questions in detail and I remain convinced that this is a good paper.

---

### Official Review · Reviewer_b4DR · 2026-03-13

**Soundness:** 4
**Presentation:** 3
**Significance:** 2
**Originality:** 2
**Overall Recommendation:** 5
**Confidence:** 4

**Summary:**

This paper proposes a novel data-dependent state duration for Bayesian nonparametric infinite hidden Markov models. Bayesian nonparametric HMMs offer the advantage of data-driven inference of models with high transparency and interpretability, including their model complexity. Previously, infinite HMMs and their semi-Markov variants have been proposed, but these models share state transition tendencies statically across the entire sequence. In contrast, real-world data often exhibit latent dynamics that are highly dependent on the observed data. This paper proposes a new model using a negative binomial distribution to represent such observation-dependent latent dynamics.

**Compliance With Llm Reviewing Policy:**

Affirmed.

**Final Justification:**

I appreciate the authors’ concise and clear response. All of my concerns and questions have been addressed. This paper offers a new and insightful perspective on modeling and theoretical analysis in the field of Bayesian nonparametric statistics. I believe this corresponds to the rating guideline “5: Technically solid paper, with high impact on at least one sub-area.” On the other hand, since the authors have (perhaps strategically) chosen to narrow the scope of this paper to the field of Bayesian nonparametric statistics rather than general sequence modeling (which includes methods based on deep learning, for example), I find it somewhat difficult to raise the score to “6” based on the current draft. However, personally, I believe the authors’ strategy of limiting the scope is very appropriate given the nature of this paper. In fact, attempting to position the methods in this paper beyond the realm of Bayesian nonparametric statistics would likely result in a very complex discussion. For these reasons, I am maintaining the score at “5,” but I would like to emphasize that this does not imply that I have any remaining concerns.

**Key Questions For Authors:**

I am deeply grateful to the authors for sharing their intriguing research ideas. Overall, this paper represents an interesting development in the field of Bayesian nonparametric research. It proposes a new model that appears highly valuable for situations where one wishes to infer data-driven, transparent, and interpretable statistical probability models. However, I have some concerns regarding the current draft and would like to comment on them.

**Q1: Positioning relative to related work**

Considering the history of Bayesian nonparametrics, I believe several studies warrant discussion of their relationship within this paper. Several Bayesian nonparametric models for segmentation have been proposed. Apart from the authors' data-dependent negative binomial distribution strategy, strategies primarily using nested structures have been employed. Discussing these relationships in this paper would further clarify the proposed method's historical context.


- Mochihashi, D., Yamada, T., & Ueda, N. (2009, August). Bayesian unsupervised word segmentation with nested Pitman-Yor language modeling. In Proceedings of the Joint Conference of the 47th Annual Meeting of the ACL and the 4th International Joint Conference on Natural Language Processing of the AFNLP (pp. 100-108).

- Mochihashi, D., & Sumita, E. (2007). The infinite Markov model. Advances in neural information processing systems, 20.

- Wood, F., Gasthaus, J., Archambeau, C., James, L., & Teh, Y. W. (2011). The sequence memoizer. Communications of the ACM, 54(2), 91-98.

- Wood, F., Archambeau, C., Gasthaus, J., James, L., & Teh, Y. W. (2009, June). A stochastic memoizer for sequence data. In Proceedings of the 26th annual international conference on machine learning (pp. 1129-1136).


I believe block diagonalization iHMMs represent a Bayesian nonparametric model that captures duration in a broad, data-dependent sense. Of course, whether BDiHMMs truly capture data-dependent duration may be subject to differing interpretations. However, the block structure learned data-driven within the state transition matrix could arguably be seen as capturing a certain type of duration “indirectly and implicitly” in a broad sense. This BDiHMM also represents one of the interesting extensions that emerged around the time HDP-HMM, HDP-HSMM, and sticky HDP-HMM were developed, making it worthy of discussion in this paper.

- Stepleton, T., Ghahramani, Z., Gordon, G., & Lee, T. S. (2009, April). The block diagonal infinite hidden Markov model. In Artificial intelligence and statistics (pp. 552-559). PMLR.


**Q2: Regarding Theoretical Analysis**

I find the analysis of finite-order truncation errors in model complexity in Section 5 to be very challenging and insightful. What is commonly seen is the consistency or convergence of posterior probabilities as data grows infinitely. Attempting to analyze finite-order truncation errors in model complexity for posterior probabilities and predictive distributions is a truly remarkable endeavor.

- First, I would like to ask the authors if there are existing papers that perform similar analyses on HDP-HMMs or stick HDP-HMMs.

- Regarding the fourth item in Step 1 of the proof of Theorem 1 in the main text (lines 610-612), does this second inequality imply that the probability of event E is higher than y' = y, even considering the situation where an emission from a state index > L miraculously ends up having an equivalent representation to an emission from a state index <= L? I apologize if this argument is unclear. My hope is that since the left-hand inequality in lines 610-612 is trivial from the Coupling Lemma, but the right-hand inequality is non-trivial, an explanation from the authors would be helpful for readers.

- I understand the intent of Theorem 5.1 in the main text. I also understand the intent of Theorem 5.4. It makes sense that you want to argue RED-HDP-HMM essentially represents a broader class of probabilistic models than HDP-HSMM. However, I don't fully grasp what the authors are trying to assert in Theorem 5.2 and Corollary 5.3. Could you explain the intent behind these two? Do these provide new insights not covered by Theorem 5.1? Perhaps, as part of Section 5's structure, providing more supplementary interpretation of the author's theoretical claims would make their intent clearer to readers. I understand space constraints, but for example, the proof sketches were not particularly helpful to me. Since the proof sketches alone didn't capture the key points well, I ended up reading the proofs in the appendix.

[Minor comment]

I believe there is room for improvement in several notations.

- In \beta \sim GEM(\gamma), \beta is an infinite-dimensional vector, yet \alpha and \gamma are used as scalars. For instance, distinguishing \beta by using boldface or similar means would make the differences clearer. Particularly, notations like DP(\alpha\beta) become very confusing because they place a scalar and a vector side by side.

- Unless I'm mistaken, Total variation distance is used as \(d_{TV}\) without being defined. If space permits, adding a note upon its first appearance could help avoid unnecessary confusion for readers.

- The notation \Pi_{HDP}^{(L)} in Theorem 1 might be changed to something like \Pi_{\infty}^{(L)}. When I first read it, I mistakenly thought \Pi_{HDP}^{(L)} referred to the conventional HDP-HMM model.

**Limitations:**

Yes. Section 9 clearly discuss its potential limitation and future direction.

**Strengths And Weaknesses:**

[Strengths]

- *Originality*: We propose a new, practically significant model for Bayesian nonparametric sequence analysis. Its structure is remarkably simple, allowing Bayesian inference algorithms to be straightforwardly extended within the existing framework. This simplicity is one of the paper's key attractions.

- *Soundness*: This paper performs a highly rigorous theoretical analysis of this simple model extension. The authors clearly demonstrate the theoretical justification for the proposed model's advantages over the conventional standard infinite hidden semi-Markov model, including the risks involved in truncating infinite complexity to finite levels.

[Weaknesses]

- *Significance*: The potential core readership of this paper may be somewhat limited to the community that heavily relies on Bayesian nonparametrics and Bayesian analysis. However, this also means it is a paper of very high value to specialists, so it may not necessarily be considered a weakness.

- *Presentation*: This paper is written in a clear and straightforward manner and is very well structured overall. However, as discussed in the questions, there appears to be room for improvement in certain sections. Additionally, the final formatting adjustments seem to have been rushed. I sincerely hope these points will be carefully addressed in the revised version.

---

> ### Author Rebuttal · Authors · 2026-03-31
>
> We sincerely thank the reviewer for their careful reading and constructive suggestions.
>
> ## Q1: Positioning Relative to Related Work
>
> **Nested BNP segmentation models.** We thank the reviewer for highlighting these important references. We agree that the broader landscape of Bayesian nonparametric sequence models deserves fuller treatment, and we will add a discussion in the Related Work section covering:
>
> - *Mochihashi et al. (2009) and Mochihashi & Sumita (2007)*: These extend BNP sequence modeling through nested Pitman-Yor language models, addressing segmentation from a language modeling perspective. While our work focuses on continuous-valued time series with explicit duration variables, both share the goal of discovering latent temporal structure without pre-specifying complexity. We will discuss how our observation-dependent duration mechanism complements the nested hierarchy approach.
>
> - *Wood et al. (2009, 2011) — Sequence Memoizer*: Uses hierarchical PY processes over unbounded context trees for discrete sequences. RED-HDP-HMM addresses continuous observations with explicit latent state structure and duration modeling. We will note this complementary relationship.
>
> **BDiHMM.** We appreciate this insightful suggestion. The block-diagonal structure in BDiHMM can be interpreted as capturing implicit, data-driven duration behavior through learned "super-states." In revision, we will note that BDiHMM (Stepleton et al., 2009) represents an intermediate approach between implicit duration models (sticky HDP-HMM) and fully explicit ones (RED-HDP-HMM). The key distinction is that RED-HDP-HMM models durations as first-class random variables with explicit observation-dependence, making the duration mechanism more directly interpretable and more flexible.
>
> ## Q2: Theoretical Analysis
>
> **Prior work on truncation error.** To our knowledge, formal truncation error bounds of the type in Theorem 5.1 have not been previously derived for HDP-HMMs. The weak-limit approximation (Ishwaran & Zarepour, 2000, 2002) was adopted by Fox et al. (2011) without explicit finite-L error bounds. The closest related analysis is Ishwaran & James (2001), who bound $L_1$ distance for truncated stick-breaking representations. Our contribution specializes this to the sequential (HMM) setting via an explicit coupling construction, where error accumulates over T time steps, yielding the $Texp(−c(L−1))$ bound. We will clarify this relationship in revision.
>
> The reviewer's reading is correct. Under our coupling: if event $E^c$ occurs (the full model never visits state $k \geq L$), then $z_t = z^‎\prime_t$ for all $t$, implying $y_t = y^‎\prime_t$. If $E$ occurs, we conservatively declare $y_{1:T} \neq y^‎\prime_{1:T}$. So $\{y_{1:T} \neq y^‎\prime_{1:T}\} \subseteq E$. The bound is conservative but clean, and the exponential decay in $L$ makes the looseness immaterial in practice. We will add this explanation to the appendix.
>
> **Intent of Theorems 5.1, 5.2, and Corollary 5.3.** In practical terms: Theorem 5.1 answers "How many states L do I need?", Theorem 5.2 answers "How large should D_max be?" (unique to explicit-duration models), and Corollary 5.3 gives total approximation quality. Theorem 5.2 bounds the error from clamping the infinite-support Negative Binomial durations at D_max — a result not needed for standard HDP-HMMs and providing independent guidance for practitioners. Corollary 5.3 combines both via the triangle inequality, showing both truncation parameters can be set independently. We will add interpretive text making this structure explicit.
>
> We take the reviewer's point on proof sketches and will replace them with clearer one-sentence summaries of the proof strategy, directing readers to the appendix.
>
> **Notation.** We will: use boldface **β** for the GEM weight vector; define $d_{TV}$ at first use; rename $\Pi_{HDP}^{(L)} \rightarrow \Pi_{\infty}^{(L)}$ to avoid ambiguity.
>
> ## Summary of Planned Revisions
>
> 1. **Related Work**: Add discussion of nested PY models, Sequence Memoizer, and BDiHMM.
> 2. **Theory section**: Add interpretive text for Theorems 5.1/5.2/Corollary 5.3; improve proof sketches; clarify coupling inequality.
> 3. **Notation**: Boldface β, define $d_{TV}$, rename $\Pi_{HDP}^{(L)} \rightarrow \Pi_{\infty}^{(L)}$.
> 4. **Formatting**: Address final issues noted by the reviewer.
>
> We are grateful for the reviewer's thorough feedback, which will significantly improve the manuscript.

---

> > ### Author Rebuttal · Reviewer_b4DR · 2026-04-03
> >
> > I appreciate the authors’ concise and clear response. All of my concerns and questions have been addressed.
> > This paper offers a new and insightful perspective on modeling and theoretical analysis in the field of Bayesian nonparametric statistics. I believe this corresponds to the rating guideline “5: Technically solid paper, with high impact on at least one sub-area.” On the other hand, since the authors have (perhaps strategically) chosen to narrow the scope of this paper to the field of Bayesian nonparametric statistics rather than general sequence modeling (which includes methods based on deep learning, for example), I find it somewhat difficult to raise the score to “6” based on the current draft. However, personally, I believe the authors’ strategy of limiting the scope is very appropriate given the nature of this paper. In fact, attempting to position the methods in this paper beyond the realm of Bayesian nonparametric statistics would likely result in a very complex discussion. For these reasons, I am maintaining the score at “5,” but I would like to emphasize that this does not imply that I have any remaining concerns.

---

### Decision · Program_Chairs · 2026-04-30

**Decision:**

Accept (spotlight)

**Comment:**

The paper is accepted: all three reviewers assigned scores of 5 (Accept) post-rebuttal.

The paper provides formal theoretical rigor and its principled resolution of a gap no prior BNP sequential model had closed. Reviewers consistently praised the model's core contribution of integrating gamma-mixed negative binomial regression into the HDP-HMM transition dynamics, which simultaneously achieves an unbounded state space, explicit non-geometric duration modeling, and observation-dependent transitions within a conjugate Gibbs framework, supported by strong empirical gains on real-world segmentation benchmarks. One reviewer noted that the paper's deliberate focus on the Bayesian nonparametric community limits its broader reach but is appropriate given the theoretical nature of the contribution; the authors are encouraged to address the notation improvements and interpretive clarifications flagged during review in the camera-ready version.